# Sylber 2.0: A Universal Syllable Embedding

## Abstract

Scaling spoken language modeling requires speech tokens that are both efficient and universal. Recent work has proposed syllables as promising speech tokens at low temporal resolution, but existing models are constrained to English and fail to capture sufficient acoustic detail. To address this, we present Sylber 2.0, a universal framework for coding speech at the syllable level, enabling efficient temporal compression and high-fidelity reconstruction across multiple languages and expressive styles. Building on the original Sylber, Sylber 2.0 improves both linguistic coverage and reconstruction quality by training on diverse multilingual speech and introducing a syllable-level acoustic encoder and vocoder. Sylber 2.0 achieves a very low token frequency around 5 Hz, while retaining both linguistic and acoustic detail. Experiments show that it performs on par with previous models operating on high-frequency baselines, and it outperforms the original Sylber by a significant margin. We further demonstrate the efficacy of Sylber 2.0 in downstream tasks, especially in English TTS and low-resource ASR. Sylber 2.0 based TTS model, SylFlow, can generate speech with competitive intelligibility and quality with current SOTA models using only 72M, and be more effective in resource-constrained ASR than previous speech coding frameworks. In sum, we establish an effective syllable-level abstraction for general spoken language. Samples can be found here: Demo link

## 1 Introduction

Modeling speech effectively requires capturing both its acoustic detail and linguistic content. Various modeling approaches have been proposed to encode such information, from acoustic spectral features to modern speech representational learning methods. Masked prediction–based speech self-supervised learning (SSL) (Hsu et al., 2021; Chen et al., 2022; Mohamed et al., 2022) has been shown to encode rich linguistic content (Pasad et al., 2021; 2023), and thus has been successful in many downstream tasks (Yang et al., 2021). However, the learned embeddings are mostly phonetic (Hsu et al., 2021; Cho et al., 2023; 2024a; Choi et al., 2024). Variational autoencoders (VAEs), with or without vector quantization (VQ), have provided compact representations of speech with audio reconstruction that can be used in generative modeling such as text-to-speech (TTS) (Défossez et al., 2022; Kumar et al., 2023; Ju et al., 2024; Ji et al., 2024; Zhang et al., 2023; Défossez et al., 2024; Guo et al., 2025; Liu et al., 2025; 2024; Wang et al., 2025a; Turetzky et al., 2024; Wu et al., 2025).

However, downstream models using speech tokens suffer from high token frequency. Unlike text, speech audio exists in continuous time without a clear delimiter, which has forced speech tokens to densely encode each frame. Some recent work has mitigated this with a large downsampling window. For example, Mimi (Défossez et al., 2024) codes speech at 12.5 Hz in quantized space. VibeVoice (Peng et al., 2025) and CLEAR (Wu et al., 2025) use VAEs to compress speech down to 7.5 Hz and 7.7 Hz, respectively. Yet, it is unknown whether these tokens, with their high compression rates and fixed windows, can encode linguistic content effectively beyond low-level acoustics, which is crucial in spoken language modeling and many other downstream tasks (Algayres et al., 2023; Cho et al., 2025; Baade et al., 2025; Yang et al., 2021).

Recent studies have proposed leveraging syllables to compress speech to around 5 Hz (Cho et al., 2025; Baade et al., 2025). These approaches are based on findings that syllabic segments naturally emerge in speech SSL without using text (Cho et al., 2024b; Komatsu & Shinozaki, 2024). By tokenizing speech using syllabic segments, (Cho et al., 2025; Baade et al., 2025) demonstrate that syllabic tokens enable more efficient spoken language modeling compared to dense tokens, with

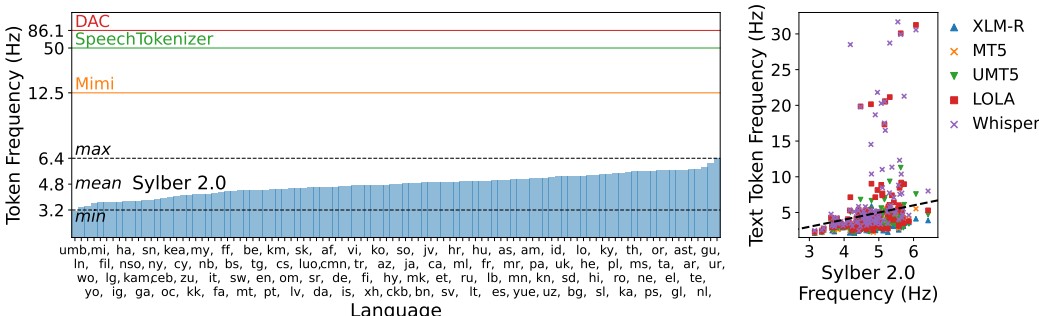

Figure 1: Comparison of token frequency of speech and text tokenization methods. (left) Each bar indicate token frequency of Sylber 2.0 for each language from 102 languages in FLEURS-R. (right) Comparison with text BPE tokens. Each dot denotes each of 102 languages.

significantly reduced token length, and show that intelligible speech can be generated from those tokens. The benefits of using syllables for speech are well supported by linguistic theories and findings in cognitive neuroscience showing that syllables are natural behavioral and cognitive units of speech (Greenberg, 1998; Oganian & Chang, 2019; MacNeilage, 1998; Greenberg, 1998). However, previous syllabic tokens were trained only on English read speech and are not generalizable to different languages and styles, which significantly limits practical utility. Moreover, those tokens severely lack acoustic detail, such that speaker identity is outsourced or completely ignored in the generative process. While the lack of acoustic information may be desirable in higher-order semantic modeling, it is nonetheless crucial for a complete speech token.

To address these issues, we propose Sylber 2.0, a universal syllabic encoding–decoding framework that can compress any arbitrary speech into syllable embeddings with a token frequency of around 5 Hz. We extend the previous SSL framework, Sylber (Cho et al., 2025), to learn syllables from diverse languages and styles. Moreover, we made several architectural changes. We introduce a boundary detector to detect syllable boundaries that arise during training, which enables faster, parallelizable segmentation. We also introduce an auxiliary acoustic encoder that encodes acoustic details of syllabic tokens, which otherwise mostly contain linguistic content. Then, we train a lightweight vocoder (Siuzdak, 2024) to synthesize the original waveform at 24 kHz from the compressed embeddings. Our experiments show that Sylber 2.0 achieves near-perfect reconstruction, closing the gap with high-frequency tokens and surpassing the original Sylber by a wide margin.

While existing speech tokenizers aim to find a minimal set of codes (analogous to "bytes" in text processing), we position Sylber 2.0 similarly to Byte Pair Encoding (BPE) + Embedding dictionary in language models, directly projecting waveforms to embeddings with variable grouping of frames. As shown in Figure 1, Sylber 2.0 achieves the lowest token frequency for speech, ranging from 3.2 Hz to 6.4 Hz across various languages, with an average of 4.8 Hz. Compared with text BPE, Sylber 2.0 shows lower frequency in less common languages (Figure 1, right). We further validate our framework by training a small TTS model which performs on-par or better than previous SOTA TTS models in terms of intelligibility and quality of generated audio.

Our contributions are summarized as follows:

- Our speech SSL framework, Sylber 2.0, can learn a universal syllabification of speech and detect syllables in many different languages.

- Sylber 2.0 can compress speech to 4.8 Hz on average across 102 languages, which is the lowest token frequency ever reported for multilingual speech.

- Sylber 2.0 outperforms the original Sylber in reconstruction quality and approximates the performance of previous high-frequency tokens, even in reconstructing expressive singing voice.

- We train a zero-shot multispeaker TTS model using Sylber 2.0 which achieves performance comparable to existing SOTA TTS models while using only 72M parameters.

- Our framework requires relatively minimal resources, fitting entirely on a single 24 GB memory GPU for training.

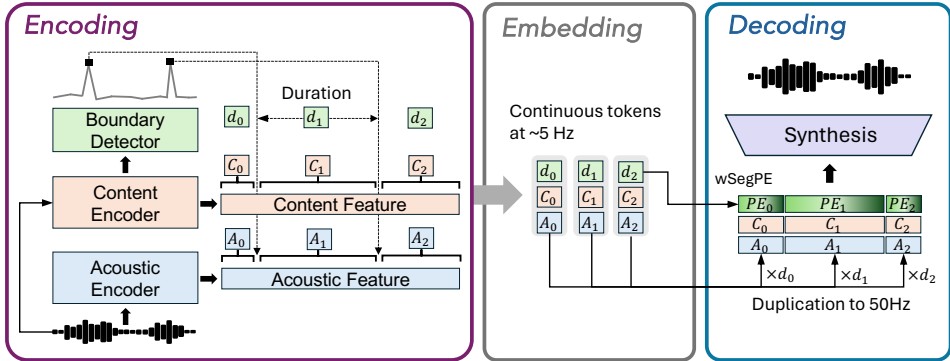

Figure 2: Encoding-decoding framework of Sylber 2.0. The model compresses speech into non-uniform ($\sim$ 5 Hz) embeddings with different components.

## 2 RELATED WORK

**Speech Tokenization** Various speech tokenization methods have been proposed to represent raw waveforms (Défossez et al., 2022; Kumar et al., 2023; Ju et al., 2024; Ji et al., 2024; Zhang et al., 2023; Défossez et al., 2024; Guo et al., 2025). Earlier works primarily focused on quantizing acoustic details, while more recent approaches incorporate additional structure such as phonetic content or disentangled speaker identity (Ju et al., 2024; Zhang et al., 2023; Défossez et al., 2024), often leveraging pretrained SSL models as guidance (e.g., by distilling SSL features). Although much of this research has concentrated on reducing the coding space, the temporal dimension remains dense. Our model addresses this gap by exploiting the emergent syllabic structure in SSL representations of speech.

Since speech tokenization is mainly used in generative modeling, high-quality decoding is essential. However, discrete approaches often rely on many codebooks, which complicates generation. To avoid this, several recent works use continuous tokens with diffusion or flow matching replacing categorical sampling in the discrete code space (Liu et al., 2025; 2024; Wang et al., 2025a; Turetzky et al., 2024; Wu et al., 2025). This simplifies decoding compared to multi-codebook generation and provides finer control over the sampling process. Because encoding at syllable frequency would otherwise require more codebooks than high-frequency tokens when quantized, we design Sylber 2.0 to operate in a continuous embedding space. This design is particularly effective for multilingual settings, where phonetic boundaries vary substantially across languages.

**Emergent Syllabic structure in speech SSL** Previous studies have demonstrated that syllables can be learned from audio without text (Cho et al., 2024b; Komatsu & Shinozaki, 2024; Baade et al., 2025; Cho et al., 2025). Cho et al. (2024b); Komatsu & Shinozaki (2024) show that self-distillation can induce syllable segments from pretrained SSL models such as HuBERT. Baade et al. (2025) leverage masked prediction loss in HuBERT to induce syllables. Then, a segmentation algorithm is applied to produce syllabic tokens at 4–5 Hz. In particular, Cho et al. (2025); Baade et al. (2025) train generative models to reconstruct intelligible speech from the syllabic tokens. However, these models lack or ignore acoustic details and are limited to audiobook-style English speech.

## 3 METHODS

### 3.1 ENCODING-DECODING FRAMEWORK WITH SYLLABIC EMBEDDINGS

Compared to previous fixed-rate speech coding, Sylber 2.0 is dynamic and flexible in producing tokens at syllabic granularity around 5 Hz. The syllabic tokens are composed of three components: *duration*, *content embedding*, and *acoustic embedding* ($d$, $C$, $A$ tokens in Figure 2). The duration indicates the length of each token, which is used to restore the original full frames for reconstruction. The content embedding represents the linguistic abstraction of the syllables that convey the intelligible content of the speech (§3.2). The acoustic embedding provides information on the acoustic details (e.g., voice identity) that are missing in the abstract content feature (§3.3). During decod-

ing, the tokens are expanded to the original frame rate using the duration information, and then a lightweight vocoder synthesizes the original waveform at a 24 kHz sampling rate. The inference pipeline is depicted in Figure 2. The model is trained with carefully curated stages of different SSL methods to learn embeddings that are linguistically grounded and compressed with extremely short token lengths. The details are explained in the following sections. Note that no text is used in any stage of training.

## 3.2 LINGUISTIC CONTENT ENCODER TRAINING

### 3.2.1 FRAME-WISE SELF-DISTILLATION

We utilize teacher–student self-distillation to induce an initial syllabic structure from a pretrained speech SSL model, where the teacher, $\mathcal{M}_T$, is the exponential moving average (EMA) of the student, $\mathcal{M}_S$. In a similar vein of vision SSL models (Chen et al., 2020; Caron et al., 2021; Oquab et al., 2023), this learning objective aims to learn invariant linguistic content from different augmented view of the input waveform, $\tau(x)$, where $x$ is waveform and $\tau \sim \mathcal{T}$ is data augmentation. More importantly, self-distillation methods can induce syllabic structure in the embedding space (Cho et al., 2024b; Komatsu & Shinozaki, 2024). In particular, we use frame-wise self-distillation, which minimizes $\text{MSE}(\mathcal{M}_S(\tau(x)), \mathcal{M}_T(\tau'(x)), \ \tau, \tau' \sim \mathcal{T}$.

We use a set of data augmentation. The speaker identity is perturbed by modifying formant levels, applied with $p = 0.3$ (Qian et al., 2022; Komatsu & Shinozaki, 2024). Environmental noise (Reddy et al., 2021) and other randomly cropped speech clips are added with $p = 0.2$ and $p = 0.05$, respectively (if applied, only one of these is chosen with equal probability). We also randomly apply room impulse responses (RIRs) sampled from the GTU-RIR corpus (Pekmezci, 2025). Lastly, random white noise is added with $p = 0.3$.

The model is initialized with a multilingual SSL model, mHuBERT (Boito et al., 2024), which is trained on 147 languages. The last three layers are randomly reinitialized. The teacher targets are L2-normalized, and student model has an additional fully-connected layer that is not used in the teacher. The EMA decay rate is set to 0.999.

### 3.2.2 SELF-SEGMENTATION DISTILLATION

The syllabic structure learned from the previous stage is further refined through self-segmentation distillation (Cho et al., 2025), an SSL method proposed in the original Sylber. The model is trained by predicting segment-averaged embeddings from the teacher, where the segments are derived by an unsupervised segmentation algorithm on the teacher's features. In particular, we minimize the Mean Squared Error between the student's outputs and the segment-averaged teacher's outputs, $\text{MSE}(\mathcal{M}_S(\tau(x)), \text{seg}(\mathcal{M}_T(x)), \ \tau \sim \mathcal{T}$ where seg means the segmentation and average pooling. The teacher is initialized with the student weights and updated through multiple stages. The same data augmentation described in §3.2.1 is applied, but only to the student's inputs.

Compared to the previous Sylber, we remove the explicit silent masking in the original Sylber to better preserve information. Sylber explicitly removed frames regarded as silent (Figure 3, top panels) by predicting "0s" for those frames. However, as a result, the model often misses syllables with low gain. Therefore, we removed this masking in our framework. Consequently, our model produces more tokens by retaining such segments, but this enables much more accurate reconstruction of the original audio.

**Boundary Detection** To replace the expensive segmentation algorithm, we introduce a boundary detector. The previous approach compared similarities between adjacent frames, incurring quadratic computational cost. Sylber introduced a greedy algorithm that reduced this to linear order, but it requires clean boundaries and is sensitive to noise as shown in Figure 3, the first and third panel. Moreover, its dynamic nature prevented parallelization in GPU. To solve this, we introduce a boundary detector to predict boundaries drawn by the unsupervised segmentation algorithm. A peak detection algorithm is then applied to the boundary probabilities, which is much faster than similarity-based algorithms. See Appendix B.1 for Real-Time Factor comparison.

**Multi-Stage Training** The training of the content encoder is divided into four stages. Except the first stage, the teacher model is initialized with the student weights at the beginning and then fixed

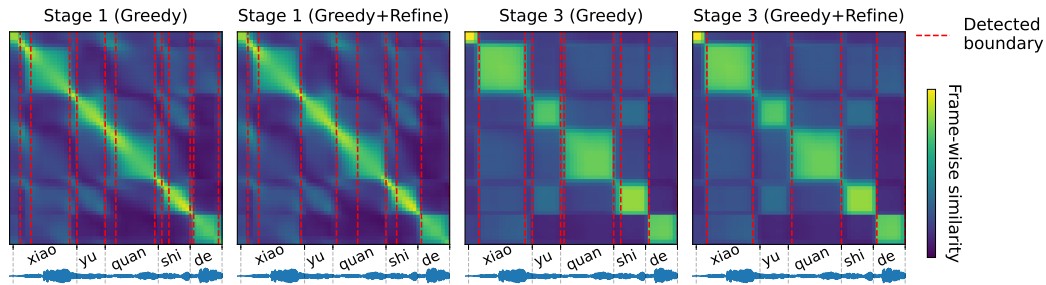

Figure 3: Similarity matrix after stage 1 and 3. The detected boundaries are denoted as red dashed lines. "Greedy+Refine" denotes using additional refinement after greedy segmentation, which can clean noisy boundaries of greedy algorithm. Note that this segmentation is replaced with the boundary detector. See Figure 4 bottom right for the final result.

throughout that stage. The stage 1 is the frame-wise self-distillation described in §3.2.1. The stage 2 and 3 follow the original Sylber, using unsupervised segmentation on the teacher's outputs to obtain target segments. We used a greedy algorithm suggested by Cho et al. (2025) with additional refinement (See Appendix A.1.1 for details). Figure 3 illustrates how the embedding space is progressively evolve into syllables after the stage 1 and 3, and the effectiveness of the refinement strategy. The last stage is trained with the boundary detector as a drop-in replacement for the segmentation algorithm. The boundary detector is trained in stage 3 and 4, using Binary Cross-Entropy to predict the probability of boundaries obtained from the teacher segments.

The student model consists of 9 Transformer layers, following Sylber. Unlike Sylber, we use the 8th layer of the teacher to extract target features. The boundary detector has 3 Transformer layers with the same architecture as the main model, followed by a fully-connected layer with a binary logit.

The trained model serves as a *content encoder* to extract the linguistic content of speech. Frames are averaged within segments predicted by the boundary detector, producing a compressed content embedding at around 5 Hz. This segment-averaged embedding is further refined with residual fully-connected layers, reducing the dimension to 64 during training of the syllable-to-speech synthesis model (see §3.4). We refer to this 64-dimensional embedding as *content embedding* ($C$ tokens in Figure 2).

### 3.3 SYLLABLE-GUIDED ACOUSTIC ENCODER

Sylber 2.0 is designed to provide a complete encoding-decoding framework that can compress speech into syllables. The previous works target single speaker generation (Baade et al., 2025) or borrow speaker embeddings from other pretrained models (Cho et al., 2025) since the acoustic information tends to be marginalized out by self-distillation (Cho et al., 2024b; Komatsu & Shinozaki, 2024; Cho et al., 2025).

We train a separate *acoustic encoder* to augment missing acoustic details learned from the self-distillation training. This additional acoustic encoding is also represented at the syllable level, by using the segments inferred by the boundary detector. The acoustic encoder consists of a CNN and 6 transformer layers. The CNN is initialized with that from WavLM-Large (Chen et al., 2022), except the 2nd layer since it has a wider (2→3) stride to increase receptive field from 320 to 480, as we use a 24 kHz input. The output frames are averaged within each segment detected from the boundary detector. Similar to the content encoder, the averaged embeddings are projected using residual fully-connected layers to a lower dimension of 64. This embedding is referred to as *acoustic embedding* of the Sylber 2.0 embedding space ($A$ tokens in Figure 2).

### 3.4 VOCODER FOR SYLLABLE-TO-SPEECH SYNTHESIS

We train a vocoder to synthesize speech from our syllabic embeddings. The content and acoustic embeddings are duplicated according to the original duration back into 50 Hz frames (Figure 2, right, *Decoding* panel). We introduce within-segment positional encoding (wSegPE), which is concatenated to each frame to indicate its relative position within the segment. The position of frame

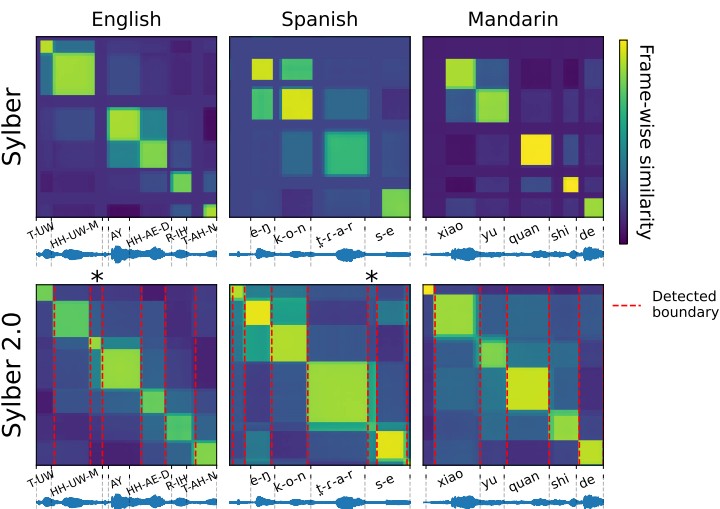

Figure 4: Similarity matrix in three different languages using Sylber (top) and Sylber 2.0 (bottom). The ground truth syllable boundaries are denoted below each plot. The boundaries detected are denoted as red dashed lines. "*" denotes the segments which are masked out in the previous Sylber.

that increases from 0 at the beginning to 1 at the end is used to lookup a learnable embedding template. Since this position is continuous, we interpolate two closest embeddings in the template. The template has 11 embeddings.

We adopt Vocos (Siuzdak, 2024) for fast, lightweight synthesis that can instantly convert syllables into speech waveforms. Our setup uses 12 ConvNext layers (Liu et al., 2022b) to predict phase and magnitude, generating 24 kHz audio through an inverse short-time Fourier transform (iSTFT). The model has only 100M parameters, making it significantly smaller than the Transformer stacks used in previous syllable-to-speech vocoders (Cho et al., 2025; Baade et al., 2025). We use the same losses as Vocos (Siuzdak, 2024), including reconstruction loss, multi-period and multi-scale adversarial losses, and feature-matching loss. In addition, we include the perceptual loss proposed by Parker et al. (2025), computed with WavLM-Large (Chen et al., 2022) using layer 0 (CNN output) and layers 3, 6, 9, and 12. An ablation experiment in Appendix B.1 well substantiates the individual contribution of the model component to the high-fidelity reconstruction.

To learn embeddings with the intended decomposition into content and acoustic embeddings, we employ some strategies during training. First, we freeze the upstream model of the content encoder and boundary detector. (The final residual FC layers in content encoder are updated). To encourage acoustic–content disentanglement, we apply random voice perturbations to both the input to the acoustic encoder and the targets, while keeping the content embedding consistent with the original speech. The acoustic embeddings within each clip are randomly averaged or shuffled across time to prevent the model from overly relying on acoustic information to generate audio.

## 3.5 TRAINING DETAILS

For training the content encoder, we use a collection of multilingual datasets including Emilia (He et al., 2024), MLS (Pratap et al., 2020), and FLEURS (Conneau et al., 2023). We exclude English and French from MLS since they are already extensively covered in Emilia. FLEURS is included to expose the model to a broader range of languages, up to 102 in total. To balance language exposure, each language in Emilia and MLS is evenly sampled during training, while FLEURS is sampled as a whole with twice the sampling probability of an individual language. To fit within a 24 GB GPU, audio is randomly cropped to 5 seconds. See Appendix A.1.3 for the hyperparameters for each training stage.

For training the acoustic encoder and synthesis model, we use a separate set of speech data with clean audio quality. Specifically, we use FLEURS-R (Ma et al., 2024), EXPRESSO (Nguyen et al., 2023), Globe (Wang et al., 2024), and GTSinger (Zhang et al., 2024). This composition covers speech with diverse styles, accents, languages, and even singing voice. During training, samples from FLEURS-R are drawn at $7\times$ the rate of the other corpora, and a random 3-second window

Table 1: Syllable detection performance measured in three languages. For Sylber 2.0, metrics are also measured with small chunks filtered. The highest scores are emphasized in **bold**.

| Model | English | | | | Spanish | | | | Mandarin | | | |
|---|---|---|---|---|---|---|---|---|---|---|---|---|
| | Pr↑ | Re↑ | F1↑ | R↑ | Pr↑ | Re↑ | F1↑ | R↑ | Pr↑ | Re↑ | F1↑ | R↑ |
| Sylber | **76.6** | 68.3 | 72.2 | 75.9 | **73.5** | 69.9 | 71.7 | 75.9 | **74.9** | 68.0 | 71.3 | 75.3 |
| Sylber 2.0 | 66.2 | **83.5** | 73.9 | 69.5 | 69.1 | **80.1** | 74.2 | 74.6 | 54.6 | **85.6** | 66.7 | 45.5 |
| filtered with ≥ 60 ms | 69.2 | 82.8 | 75.4 | 73.9 | 70.5 | 79.5 | 74.7 | 76.1 | 60.3 | 84.9 | 70.5 | 58.4 |
| ≥ 80 ms | 71.7 | 81.7 | **76.3** | 77.1 | 71.7 | 78.5 | **74.9** | 77.3 | 65.9 | 83.5 | 73.7 | 69.1 |
| ≥ 100 ms | 74.1 | 76.3 | 75.2 | **78.6** | 72.3 | 72.8 | 72.5 | 76.5 | 69.6 | 81.5 | 75.1 | 74.8 |
| ≥ 120 ms | 74.6 | 70.5 | 72.5 | 76.5 | 71.7 | 66.0 | 68.7 | 73.3 | 71.2 | 79.4 | **75.1** | **76.8** |

is cropped. We train the model with multiple cycles of learning rate schedules, where the training strategies described in §3.4 are differentially applied in each cycle; see Appendix A.1.4 for details.

Sylber 2.0 training requires minimal computational resources, as each stage can be run on a single NVIDIA RTX A5000 GPU with 24 GB of memory. This is a significant advantage compared to modern large-scale speech model training.

## 4 EMERGENT MULTILINGUAL SYLLABIC STRUCTURE

As we can see in Figure 4, Sylber 2.0 successfully learns syllabic structure without any textual supervision. In the similarity matrices shown in Figure 4, the detected boundaries are well aligned with syllable boundaries inferred from transcripts. The embeddings within segments are highly consistent, forming flat "squares" between boundaries. We evaluate syllable detection performance on three languages: English, Spanish, and Mandarin that were used in (Cho et al., 2025). We measure precision (Pr), recall (Re), F1 score (F1), and R-value with a 50 ms tolerance from the text-based syllable boundaries. The test data are taken from LibriSpeech (Panayotov et al., 2015), MLS (Pratap et al., 2020), and AISHELL-3 (Shi et al., 2021) for each respective language.

We compare Sylber 2.0 syllable detection with the original Sylber. Since Sylber 2.0 preserves small silent tokens, we also report scores after filtering out short segments with thresholds ranging from 60 ms to 120 ms (Table 1). As shown in the table, Sylber 2.0 achieves significantly higher recall and lower precision across all three languages, resulting in higher F1 scores for English and Spanish. This suggests that Sylber 2.0 more accurately recovers text-driven syllable boundaries, albeit with additional short segments (denoted as * in Figure 4). These extra segments can be removed through threshold-based filtering, boosting R-values beyond those of the original Sylber. Since our goal is speech coding with minimum loss, we retain all segments without filtering in our framework.

## 5 HIGH-FIDELITY, NEAR-LOSSLESS COMPRESSION AT 5 HZ

To evaluate the reconstruction performance of Sylber 2.0, we measure both intelligibility and quality of resynthesized audio. For intelligibility, we use Whisper-Large-v3 (Radford et al., 2023) to transcribe audio and compute the word error rate (WER), and we also measure short-time objective intelligibility (STOI) (Taal et al., 2010). For perceptual quality, we use Perceptual Evaluation of Speech Quality (PESQ) and UTMOS (Saeki et al., 2022).

We measure these metrics on the test sets of three corpora: LibriTTS (Zen et al., 2019), FLEURS-R (Ma et al., 2024), and GTSinger (Zhang et al., 2024). For LibriTTS, we report scores separately for the clean and other subsets. For FLEURS-R, we target 20 languages with sufficiently low Whisper transcription error rates,[1] and provide individual scores for representative languages. For GTSinger, we measure F0 (pitch) reconstruction instead of WER, since pitch is more crucial in singing. We use CREPE (Kim et al., 2018) to extract F0 and report the Pearson correlation coefficient (F0-PCC) and coefficient of determination (F0-$R^2$) for voiced frames.

---

[1] Selected languages: ko, ja, es, it, cmn, pt, de, ca, en, fr, pl, nl, ru, tr, uk, id, nb, sv, fi, ms.

Table 2: Resynthesis performance on different datasets. The highest scores are emphasized in **bold**, separately for low-frequency regime (Sylber and Sylber 2.0) and high-frequency regime (others). See Appendix 9 for full FLEURS-R scores.

**English (LibriTTS)**

| | | test-clean | | | | | test-other | | | | |
|---|---|---|---|---|---|---|---|---|---|---|---|
| Model | Hz | WER↓ | STOI↑ | PESQ↑ | UTMOS↑ | SSIM↑ | WER↓ | PESQ↑ | STOI↑ | UTMOS↑ | SSIM↑ |
| DAC | 86.1 | **3.32** | **0.99** | **4.46** | 3.92 | **1.00** | **5.99** | **0.99** | 4.43 | 3.40 | **1.00** |
| FACodec | 80 | 3.49 | 0.95 | 2.91 | **4.04** | 0.97 | 7.14 | 0.93 | 2.61 | **3.48** | 0.96 |
| SpeechTokenizer | 50 | 3.53 | 0.93 | 2.61 | 3.80 | 0.96 | 7.81 | 0.90 | 2.41 | 3.28 | 0.95 |
| WavTokenizer | 40 | 16.78 | 0.87 | 1.79 | 3.51 | 0.88 | 31.04 | 0.84 | 1.69 | 3.08 | 0.87 |
| Mimi | 12.5 | 3.59 | 0.97 | 3.47 | 3.85 | 0.97 | 7.06 | 0.95 | 3.25 | 3.33 | 0.97 |
| Sylber | 4.22 | 5.44 | 0.75 | 1.13 | **4.09** | 0.76 | 13.29 | 0.72 | 1.13 | **3.91** | 0.71 |
| Sylber 2.0 (Ours) | 5.81 | **3.86** | **0.89** | **1.99** | 3.80 | **0.92** | **8.58** | **0.87** | 1.89 | 3.54 | **0.91** |

**Multilingual Speech (FLEURS-R)**

| | | Spanish | | | | | French | | | | | Russian | | | | |
|---|---|---|---|---|---|---|---|---|---|---|---|---|---|---|---|---|
| Model | Hz | WER↓ | STOI↑ | PESQ↑ | UTMOS↑ | SSIM↑ | WER↓ | PESQ↑ | STOI↑ | UTMOS↑ | SSIM↑ | WER↓ | PESQ↑ | STOI↑ | UTMOS↑ | SSIM↑ |
| DAC | 86.1 | **2.91** | **1.00** | **4.53** | 3.29 | **1.00** | **6.34** | **1.00** | **4.52** | 3.08 | **1.00** | **5.26** | **1.00** | **4.51** | 3.32 | **1.00** |
| FACodec | 80 | 3.21 | 0.96 | 3.34 | **3.32** | 0.98 | 8.90 | 0.95 | 2.90 | **3.21** | 0.98 | 5.88 | 0.95 | 2.86 | **3.34** | 0.98 |
| SpeechTokenizer | 50 | 3.13 | 0.94 | 3.06 | 2.96 | 0.97 | 8.71 | 0.92 | 2.72 | 2.91 | 0.97 | 6.02 | 0.93 | 2.63 | 3.04 | 0.98 |
| WavTokenizer | 40 | 14.57 | 0.87 | 1.98 | 2.73 | 0.92 | 53.89 | 0.85 | 1.81 | 2.89 | 0.90 | 27.58 | 0.85 | 1.72 | 2.73 | 0.90 |
| Mimi | 12.5 | 2.93 | 0.98 | 3.91 | 3.14 | 0.98 | 6.66 | 0.96 | 3.64 | 2.98 | 0.98 | 5.45 | 0.97 | 3.56 | 3.15 | 0.98 |
| Sylber | 3.71 | 10.66 | 0.77 | 1.18 | **3.37** | 0.84 | 59.03 | 0.74 | 1.22 | **3.61** | 0.82 | 24.13 | 0.75 | 1.19 | **3.56** | 0.80 |
| Sylber 2.0 (Ours) | 4.86 | **3.18** | **0.93** | **2.56** | 2.91 | **0.98** | **8.92** | **0.91** | **2.28** | 2.99 | **0.97** | **6.57** | **0.91** | **2.23** | 3.16 | **0.98** |

| | | Mandarin | | | | | Korean | | | | | 20 Languages | | | | |
|---|---|---|---|---|---|---|---|---|---|---|---|---|---|---|---|---|
| Model | Hz | WER↓ | STOI↑ | PESQ↑ | UTMOS↑ | SSIM↑ | WER↓ | PESQ↑ | STOI↑ | UTMOS↑ | SSIM↑ | WER↓ | PESQ↑ | STOI↑ | UTMOS↑ | SSIM↑ |
| DAC | 86.1 | **6.93** | **1.00** | **4.53** | 3.16 | **1.00** | **4.30** | **1.00** | **4.52** | 3.51 | **1.00** | **6.03** | **1.00** | **4.52** | 3.29 | **1.00** |
| FACodec | 80 | 8.65 | 0.95 | 3.02 | **3.19** | 0.98 | 5.36 | 0.96 | 3.24 | **3.63** | 0.98 | 7.29 | 0.95 | 3.06 | **3.35** | 0.98 |
| SpeechTokenizer | 50 | 8.48 | 0.93 | 2.80 | 2.91 | 0.97 | 4.96 | 0.94 | 2.96 | 3.25 | 0.97 | 7.59 | 0.93 | 2.81 | 3.06 | 0.97 |
| WavTokenizer | 40 | 38.97 | 0.86 | 1.80 | 2.70 | 0.89 | 24.42 | 0.88 | 1.97 | 2.97 | 0.90 | 33.94 | 0.86 | 1.82 | 2.79 | 0.90 |
| Mimi | 12.5 | 7.39 | 0.97 | 3.67 | 3.00 | 0.98 | 4.69 | 0.98 | 3.78 | 3.38 | 0.98 | 6.35 | 0.97 | 3.72 | 3.17 | 0.98 |
| Sylber | 3.71 | 38.10 | 0.75 | 1.21 | **3.62** | 0.83 | 17.96 | 0.80 | 1.22 | **3.61** | 0.82 | 28.42 | 0.76 | 1.18 | **3.55** | 0.82 |
| Sylber 2.0 (Ours) | 4.86 | **8.21** | **0.91** | **2.29** | 2.96 | **0.97** | **5.05** | **0.94** | **2.60** | 3.30 | **0.97** | **7.57** | **0.92** | **2.35** | 3.09 | **0.98** |

**Singing Voice (GTSinger)**

| Model | Hz | F0-PCC(r)↑ | F0-$R^2$↑ | STOI↑ | PESQ↑ | UTMOS↑ | SSIM↑ |
|---|---|---|---|---|---|---|---|
| DAC | 86.1 | **0.99** | **0.99** | **0.96** | **4.37** | 2.43 | **1.00** |
| FACodec | 80 | 0.97 | 0.85 | 0.84 | 2.85 | **2.46** | 0.98 |
| SpeechTokenizer | 50 | 0.97 | 0.84 | 0.77 | 2.35 | 2.15 | 0.96 |
| WavTokenizer | 40 | 0.94 | 0.76 | 0.70 | 1.81 | 2.07 | 0.92 |
| Mimi | 12.5 | 0.98 | 0.95 | 0.86 | 3.26 | 2.29 | 0.97 |
| Sylber | 1.87 | 0.78 | -2.41 | 0.47 | 1.11 | 2.23 | 0.80 |
| Sylber 2.0 (Ours) | 3.37 | **0.96** | **0.88** | **0.73** | 2.14 | **2.33** | **0.95** |

Additionally, we evaluate speaker similarity (SSIM) through cosine similarity using Resemblyzer speaker embeddings.[2] We compare Sylber 2.0 against several representative open-sourced speech tokenizers operating at different token frequencies (12.5–86.1 Hz): DAC(Kumar et al., 2023), FA-Codec (Ju et al., 2024), SpeechTokenizer (Zhang et al., 2023), WavTokenizer (Ji et al., 2024), and Mimi (Défossez et al., 2024), along with the original Sylber.

As shown in Table 9, Sylber 2.0 reconstructs high-quality audio that preserves intelligible content. Sylber 2.0 outperforms Sylber by a wide margin in every aspect except UTMOS. In fact, some UTMOS scores of Sylber are even higher than those of high-frequency tokens, because Sylber's resynthesis relies on an external vocoder with a quality-enhancement capacity (Cho et al., 2024c).

Several Sylber 2.0 scores approach those of high-frequency tokens. Specifically, WER scores show only minimal gaps: Sylber 2.0 achieves 7.57% WER across 20 languages in FLEURS-R, while the best high-frequency token, DAC, achieves 6%, a trend also reflected in individual languages and corpora. PESQ scores are reasonably high, but below those of high-frequency tokens. This is because that PESQ is sensitive to frame-level acoustic details that may be lost when averaging within

---

[2]https://github.com/resemble-ai/Resemblyzer

Table 3: TTS performance on LibriSpeech (PC) test-clean and SeedTTS English test set.

| Model | #Params | Training Data | LibriSpeech-PC | | | SeedTTS English | | |
|---|---|---|---|---|---|---|---|---|
| | | | WER(%)↓ | SIM-o↑ | UTMOS↑ | WER(%)↓ | SIM-o↑ | UTMOS↑ |
| Ground Truth | - | - | 2.47 | 0.69 | 4.09 | 2.14 | 0.73 | - |
| CosyVoice (Du et al., 2024a) | 300M | Multi-170k | 3.59 | 0.66 | - | 4.08 | 0.64 | - |
| CosyVoice 2 (Du et al., 2024b) | 500M | Multi-170k | 2.47 | 0.65 | **4.35** | 2.57 | 0.65 | - |
| FireRedTTS (Guo et al., 2024) | 580M | Multi-248k | 2.69 | 0.47 | - | 3.82 | 0.46 | - |
| MaskGCT (Wang et al., 2025c) | 1048M | Emilia-100k | 2.72 | **0.69** | 3.90 | 2.62 | 0.71 | - |
| F5-TTS (Chen et al., 2024) | 300M | Emilia-100k | 2.42 | 0.66 | 3.88 | **1.83** | 0.65 | - |
| DiTAR (Jia et al., 2025) | 600M | Emilia-100k | 2.39 | 0.67 | 4.22 | 1.69 | **0.74** | - |
| SparkTTS (Wang et al., 2025b) | 500M | Multi-100k | - | - | 4.35 | 1.98 | 0.58 | - |
| CLEAR-Base (Wu et al., 2025) | 439M | Libri-50k | 2.21 | 0.59 | 4.22 | - | - | - |
| CLEAR-Large (Wu et al., 2025) | 686M | Libri-50k | **1.88** | 0.59 | 4.22 | - | - | - |
| SylFlow (Ours) | 72M | LibriTTS-0.6k | 3.10 | 0.31 | 4.27 | 2.62 | 0.31 | 4.19 |
| | | +Emilia-47k | 2.35 | 0.36 | 4.33 | 1.92 | 0.35 | 4.31 |
| *Ablation*–Replacing Sylber 2.0 | | | | | | | | |
| with Mel | 109M | LibriTTS-0.6k | 5.73 | 0.18 | 3.28 | 4.48 | 0.18 | 3.38 |
| with Mimi | 74M | LibriTTS-0.6k | 10.36 | 0.30 | 2.38 | 8.21 | 0.31 | 2.49 |

segments. For singing voice reconstruction, Sylber 2.0 shows F0 correlations comparable to high-frequency tokens, with $R^2$ values even surpassing some of them. This high level of reconstruction quality is achieved with a very low token frequency. While it can vary across languages and styles, Sylber 2.0 averages around 5 Hz: 4.8 Hz across 102 languages in FLEURS-R (Figure 1, left; a full report is in Table 12). Interestingly, Sylber 2.0 can be even more efficient in low-resource languages, where text BPE token frequencies (Conneau et al., 2020; Xue et al., 2021; Chung et al., 2023; Srivastava et al., 2025; Radford et al., 2023) are significantly higher (Figure 1, right).

Sylber 2.0 may adopt a universal rule applicable to many languages, even if it does not align precisely with language-specific syllabification rules defined by linguists. Given the lack of consensus on syllabification rules in linguistics (Anderson, 1985; Goldsmith et al., 2011; Treiman & Danis, 1988), our method provides *a universal syllabification that is consistent across languages* that is naturally emerging from machine speech perception. The extensive analysis on multiple languages and styles supports this idea.

# 6 DOWNSTREAM APPLICATIONS

## 6.1 TEXT-TO-SPEECH THROUGH SYLLABLE EMBEDDING

To further demonstrate downstream utility, we train a small text-to-speech (TTS) model using Sylber 2.0 embeddings. Since the embedding conveys full speech information, our goal is to generate these embeddings directly from text. As Sylber 2.0 tokenizes speech in a continuous space, we adopt a continuous-value autoregressive (AR) model (Liu et al., 2025; 2024; Wang et al., 2025a; Turetzky et al., 2024; Wu et al., 2025). Inspired by Wu et al. (2025), we use rectified flow (RF) (Liu et al., 2022a) with AR backbone, which we call *SylFlow*. We conduct a controlled experiments using relatively small amount of data (LibriTTS; 560 hours). We train small models with 72-109M parameters for prompting and generating Sylber 2.0, and Mel spectrogram (Mel) and Mimi for the baselines. We specifically choose these two baselines as Mimi has the shortest token length (12.5 Hz) among the baseline tokens, and Mel spectrogram can set a reference performance without any learned linguistic feature. Additionally, we train the same model with additional 47k hours of training data from Emilia (He et al., 2024). We evaluate on the LibriSpeech (PC) test-clean subset (Chen et al., 2024) and SeedTTS English test set (Anastassiou et al., 2024), where we measured WER, speaker similarity (Sim-o) and UTMOS. See Appendix A.2 for details.

As shown in Table 3, Sylber 2.0 outperforms the baseline speech representations, Mel and Mimi, in all metrics and test sets. This indicates that Sylber 2.0 may encode information in a better abstract format than dense tokens or raw acoustics, especially for generative modeling. This is well-aligned with the Sylber 2.0's correspondence to the syllables which are by definition the units of speech production. Therefore, Sylber 2.0 can serve as a more natural medium for generative speech modeling. Moreover, by scaling the amount of training data, our model can close the performance gap from

the SOTA TTS model especially on the WER (1.92-2.35 %) and UTMOS (4.31-4.33) scores, while being 5-10 × smaller in size. This suggests that Sylber 2.0 provides a more efficient and effective alternative to previous speech tokens. However, the gap in speaker similarity still remains large, which could potentially be resolved by scaling model size. As our main focus is more on syllable embeddings, we leave this as a future direction.

## 6.2 LOW-RESOURCE ASR

To further assess Sylber 2.0 embeddings, we train shallow ASR models for four different languages–English, Korean, Bemba, and Quecha–in a low-resource setting. The former two are chosen as representative languages as English is a dominant proportion in the training set, and Korean is highly syllabic, thus it would be interesting to see the link to the syllabic speech embedding. We use the 100 hour subset of LibriSpeech for training and test-clean for evaluation. For Korean we use Zeroth corpus (Jo & Lee, 2018) that contains 51.6 hours for training and 1.2 hours for testing. Bemba and Quecha are the actual low-resource languages, which are spoken only by 10 and 8 million people, respectively. These two languages are not involved in any training stage, including mHuBERT pretraining. For Bemba, we use 20 hours and 2 hours of train and test data released by (Sikasote & Anastasopoulos, 2022). For Quecha, we use the data by Sicherman & Adi (2023), where we use 48 hours for training and 1 hour for testing. For the ASR architecture, we adopt RNN-T (Graves, 2012b) as it offers more flexibility in input token length.[3] See Appendix A.3 for details.

Table 4: Performance on low-resource ASR.

| Model | en | | ko | | bem | | que | |
| --- | --- | --- | --- | --- | --- | --- | --- | --- |
| | CER↓ | WER↓ | CER↓ | WER↓ | CER↓ | WER↓ | CER↓ | WER↓ |
| Mel | 8.7 | 20.6 | 10.6 | 16.0 | 19.4 | 61.3 | 31.4 | 66.8 |
| DAC | 36.2 | 59.3 | 100.3 | 125.3 | 30.4 | 79.5 | 51.1 | 87.3 |
| Mimi | 17.1 | 32.8 | 20.9 | 31.6 | 29.6 | 80.1 | 39.9 | 76.8 |
| Sylber | **7.8** | **16.2** | 22.0 | 28.5 | 23.6 | 67.4 | 44.2 | 83.6 |
| Sylber 2.0 | 11.7 | 24.2 | **7.2** | **9.4** | **12.1** | **47.4** | **30.1** | **66.4** |
| Sylber 2.0-*Content only* | 12.1 | 24.7 | 7.3 | 9.6 | 12.7 | 48.5 | 32.1 | 69.8 |
| Sylber 2.0-*Acoustic only* | 31.5 | 52.9 | 22.3 | 32.4 | 39.5 | 86.6 | 48.8 | 86.1 |

As shown in Table 4, Sylber 2.0 enables more accurate recognition in Korean, Bemba, and Quecha. For English, the previous Sylber demonstrates lower error rates, which could be natural as it is only trained on LibriSpeech, thus potentially more fine-tuned to the domain. In fact, the performance leaps are significant in other three languages. In particular, Korean shows the lowest error rates, which may be benefitted by the the syllabic nature of the language. Sylber 2.0 shows better performance than previous tokesn, DAC and Mimi, in all languages which indicates Sylber 2.0 embeddings are closer to text than dense tokens. We also train models using the content feature or acoustic feature only to show the source of information in our decomposed embeddings. The models show minimal degradation in performance when using content feature, while using acoustic feature induces severe increases in errors, suggesting that the content feature is major information source. In fact, the content feature is phonetically organized in the embedding space (Appendix Figure 5). See Appendix B.1 for additional downstream evaluation on SUPERB benchmark.

## 7 CONCLUSION

We propose a novel speech embedding model, Sylber 2.0, that learns a universal syllabification of raw speech audio. Sylber 2.0 can compress speech from any language, style, or even singing into a linguistically grounded token sequence at around 5 Hz. Its decoder can reconstruct the original audio nearly perfectly from this short sequence, surpassing the original Sylber. Sylber 2.0 offers strong potential for transparent and efficient speech tokenization, as well as for scalable and effective spoken language modeling.

---

[3]The common practice of using CTC (Graves, 2012a) is not applicable as Sylber 2.0 often produces shorter tokens than the target character length.

ETHICS STATEMENT

We view our model as a significant advancement in speech modeling and spoken language understanding. Our approach provides efficient and effective speech tokenization, though it also carries the potential for misuse. Therefore, it is essential that users, researchers, and developers apply this model and framework with ethical care and responsibility.

REPRODUCIBILITY STATEMENT

In keeping with the principles of open research, we will make all code related publicly available. This includes both the pretrained model weights and the resources needed to retrain the model.

USE OF LLM TECHNOLOGY

In this research project, standard LLMs were used for general assistance such as for formatting and styling tables or with basic tab-completion for coding.

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

# A APPENDIX

## A.1 ADDITIONAL IMPLEMENTATION DETAILS

### A.1.1 GREEDY SEGMENTATION WITH REFINEMENT FOR STAGE-2 AND -3 TRAINING

The greedy algorithm for unsupervised segmentation proposed by Cho et al. (2025) segments frames by grouping adjacent frames with cosine similarity above a threshold ("merge threshold") in a single sweep. However, this requires embeddings with clean boundaries, which are not available in the early stages of training. To address this, Cho et al. (2025) used pre-extracted segments from a clustering-based algorithm for the first stage of training. At the same time, the authors reported that boundary errors are not critical for syllable learning; rather, the granularity of the initial segments is more important (see Appendix A.2.6 of Cho et al. (2025)).Therefore, we merge segments shorter than 80 ms that have cosine similarity with adjacent segments higher than a "refinement threshold." This preserves syllable-level token frequency while maintaining the linear computational complexity. Figure 3 illustrates the effectiveness of refinement (right and middle plots).

### A.1.2 THRESHOLD SETTING

The merge threshold is sampled from [0.5, 0.7] in the first stage and [0.7, 0.9] in the second stage, increasing sensitivity over time. Similarly, the refinement threshold is set to 0.5 for the first stage and 0.7 for the second.

For the peak detection in the boundary detection, we set a minimum peak value of 0.2, filtering boundaries with either prominence $> 0.05$ or probability $> 0.8$ (prominence of 0.1 during inference yields minimal difference).

### A.1.3 OPTIMIZER CONFIGURATIONS USED IN CONTENT ENCODER

The content encoder training is comprised of four stages. Table 5 denotes the hyperparameters used in the AdamW optimizer for each training stage.

### A.1.4 OPTIMIZER CONFIGURATIONS USED IN ACOUSTIC ENCODER AND SYNTHESIS MODEL

The acoustic encoder and synthesis model are trained jointly through multiple cycles of cosine learning rate schedule. All cycles use a batch size of 12 with $\beta_1 = 0.8$ and $\beta_2 = 0.9$ with weight decay of 0.01 for the AdamW optimizer. The number of iterations and the training strategies described in §3.4 are differentially applied to each cycle. Table 6 denotes the information.

Table 5: Hyperparameters used in each training stage.

| Stage | Batch size | Learning rate | Warmup steps | $\beta_1$ | $\beta_1$ | Weight decay | Iterations |
|---|---|---|---|---|---|---|---|
| Stage-1 | 72 | 1e-4 | 2000 | 0.9 | 0.999 | 1e-3 | 100K |
| Stage-2 | 50 | 1e-4 | 1000 | 0.9 | 0.95 | 1e-2 | 100K |
| Stage-3 | 50 | 1e-4 | 1000 | 0.9 | 0.95 | 1e-2 | 100K |
| Stage-4 | 50 | 1e-5 | 1000 | 0.9 | 0.95 | 1e-2 | 200K |

Table 6: Hyperparameters used in each cycle.

| Cycle | Max learning rate | Iterations | Perceptual Loss | Freeze Content Encoder | Freeze Acoustic Encoder |
|---|---|---|---|---|---|
| Cycle-1 | 5e-5 | 2000K | no | yes | no |
| Cycle-2 | 1e-5 | 2000K | yes | yes | no |
| Cycle-3 | 1e-5 | 100K | yes | yes | no |
| Cycle-4 | 1e-5 | 100K | yes | yes | yes |

| Cycle | Perturbing Voice Prob. | Perturbing Audio Prob. | Mean-pooling Acoustics Prob. | Shuffling Acoustics Prob. |
|---|---|---|---|---|
| Cycle-1 | 0.2 | 0.2 | 0.2 | 0.0 |
| Cycle-2 | 0.2 | 0.2 | 0.1 | 0.0 |
| Cycle-3 | 0.2 | 0.2 | 0.1 | 0.5 |
| Cycle-4 | 0.0 | 0.0 | 0.0 | 0.0 |

## A.2 TTS EXPERIMENT DETAILS

### A.2.1 IMPLEMENTATION DETAILS OF SYLFLOW

The AR backbone consists of 12 causal Transformer layers with 12 heads and hidden dimension of 512, a relatively small model size. The RF head comprises 6 residual fully-connected layers with hidden dimension of 512.

Following recent zero-shot AR TTS approaches, the TTS input consists of Sylber 2.0 tokens for style prompts and phoneme tokens from text, wrapped with special tokens at each end. The model is trained to generate the next Sylber 2.0 tokens. During training, style tokens are randomly selected from other clips of the same speaker, cropped at random with a length ratio in [0.1, 1.0]. We use the noise schedule from (Wu et al., 2025), which samples more time points in the middle. We also adopt the auxiliary velocity direction loss based on cosine distance (Wu et al., 2025). Finally, a separate end predictor is trained as a binary classifier to predict whether a given token is the last in the sequence.

### A.2.2 IMPLEMENTATION DETAILS OF MEL AND MIMI BASELINES

We use the encoder and decoder of Mimi and trained the same TTS model switching the Sylber 2.0 embedding with Mimi embedding. For the Mel spectrogram, simply switching target embedding with a dense frame rate (50Hz) was not successful. Thus, we have slightly modified the architecture. In particular, we chunk every 10 frames which are weighted averaged using a convolutional layer before fed to the LM backbone. The rectified flow is then generating the whole 10 frames at once from the next token embedding from the LM. This chunking also makes the unit of LM operation at 5Hz, matching the temporal resolution of Sylber 2.0, thus providing a more fair comparison. We use a pretrained Mel vocoder by Du et al. (2024b). All models are trained for 1M iterations and each iteration includes 2000 tokens. All models use 10 steps for generating each token at RF head.

Table 7: Real-time factor

| Batchsize | Model | Content↓ | Encoding RTF Segmentation↓ | Acoustic↓ | Total↓ | Decoding RTF↓ | E2E RTF↓ |
|---|---|---|---|---|---|---|---|
| 1 | mHuBERT | – | – | – | 0.00213 | – | – |
| | WavLM-Large | – | – | – | 0.00404 | – | – |
| | DAC | – | – | – | 0.00540 | 0.00212 | 0.00752 |
| | Mimi | – | – | – | 0.00441 | 0.00202 | 0.00643 |
| | Sylber | 0.00184 | 0.00344 | 0.01290 | 0.01818 | 0.01800 | 0.03618 |
| | Sylber 2.0 | 0.00211 | 0.00290 | 0.00269 | 0.00769 | 0.00165 | 0.00935 |
| 32 | mHuBERT | – | – | – | 0.00027 | – | – |
| | WavLM-Large | – | – | – | 0.00056 | – | – |
| | DAC | – | – | – | 0.00379 | 0.00635 | 0.01014 |
| | Mimi | – | – | – | 0.00076 | 0.00128 | 0.00204 |
| | Sylber | 0.00027 | 0.00204 | 0.09979 | 0.10210 | 0.01089 | 0.11299 |
| | Sylber 2.0 | 0.00029 | 0.00142 | 0.00144 | 0.00315 | 0.00158 | 0.00473 |

### A.2.3 EVALUATION METRICS OF TTS

We measure WER using Faster-Whisper-Large-v3 (Radford et al., 2023) for English and Paraformer-zh (Gao et al., 2023) for Mandarin, speaker similarity (SIM-o) using WavLM-TDNN Chen et al. (2022), and UTMOS Saeki et al. (2022).

### A.3 ASR EXPERIMENT DETAILS

The RNN-T (Graves, 2012b) is comprised with an encoder (transcriber), autoregressive language model (predictor), and joiner. For transcriber, We use 4 layers of bidirectional LSTM with 1024 hidden dims for each direction with dropout rate of 0.05 for feed-forward processing of given speech embedding. For the RNN-T predictor, we use 3 layers of unidirectional LSTM with 1024 hidden dims with dropout rate of 0.05. We use Torch implementation of RNNTLoss and RNNTBeamSearch for training and decoding speech. The model is trained with AdamW optimizer with learning rate linearly ramped up to 0.001 for first 500 steps and annealed to 0 using cosine schedule through 100k steps. The batch size varies from 3 to 8 to fit the training in 49GB GPU memory.

## B ADDITIONAL ANALYSES

### B.1 RTF COMPUTATION

Table 7 shows a detailed decomposition of real-time factor (RTF). In addition to Sylber, we included DAC and Mimi for dense tokenizations and mHuBERT and WavLM-Large for as reference of general speech encoding models. Compared to Sylber, Sylber 2.0 outperforms in both encoding and decoding since Sylber leverages a slow off-the-shelf speaker model and generates through multiple iterations of conditional flow-matching model. As our model has an ad hoc segmentation algorithm and non-uniform averaging/expanding mechanisms, Sylber 2.0 is slower in encoding than the straight feed-forward models, mHuBERt and WavLM-Large. However, the gap is narrower compared to Mimi or DAC, except in the batch case of DAC which is slower than Sylber 2.0. Moreover, the speed of decoding is faster or compatible with Mimi or DAC. This indicates that Sylber 2.0 can be utilized without heavy computations despite the complicated architecture.

## B.2 SUPERB BENCHMARK

We evaluate Sylber 2.0 on the tasks from SUPERB benchmark (Yang et al., 2021). It is not common practice for speech tokenization model but we provide this evaluation to provide a broader sense of information encoded by Sylber 2.0. We specifically compare performance against Mimi and Sylber (Table 8). Sylber 2.0 shows similar performance as Sylber but shows significantly better performance in speaker-related tasks: SID, ASV, and SD. Being language universal is a potential source of performance degradation in other tasks as they are all in English, though the degree is marginal. In all metrics, Sylber 2.0 outperforms Mimi.

Table 8: SUPERB Benchmark

| Model | PR | KS | IC | SID | ER | ASR | ASR (w/ LM) | QbE | SF | | ASV | SD |
|---|---|---|---|---|---|---|---|---|---|---|---|---|
| | PER↓ | Acc↑ | Acc↑ | Acc↑ | Acc↑ | WER↓ | WER↓ | MTWV ↑ | F1↑ | CER↓ | EER↓ | DER↓ |
| Mimi | 39.79 | 93.12 | 90.61 | 34.23 | 56.05 | 91.04 | 88.65 | 0.0108 | 6.72 | 98.86 | 12.17 | 13.37 |
| Sylber | 88.79 | **97.11** | **99.08** | 51.25 | **65.25** | **12.04** | **8.88** | **0.0591** | 85.66 | **29.49** | 8.75 | 15.55 |
| Sylber 2.0 | **14.74** | 96.85 | 98.05 | **72.03** | 62.37 | 14.23 | 10.04 | 0.0471 | **85.94** | 30.09 | **5.69** | **6.10** |

## B.3 ABLATION EXPERIMENT

To investigate the contribution of each component, we trained following ablation conditions for training decoder: 1) without wSegPE, 2) without Acoustic Encoder, and 3) replacing Sylber 2.0 content encoder with Sylber 1's segments/embedding. When using Sylber 1, we also trained the same Acoustic Encoder and wSegPE for its own decoder. Due to the limited time line, we limit the training data to be only FLEURS-R. For a fair comparison, we also retrained the main full model as well. Similar to the main analysis, we evaluated the reconstruction performance on FLEURS-R and also on the singing data corpus, GTSinger. This dataset is suitable to see the capability of out-of-distribution (OOD) generalization since the syllable durations in singing are significantly different from regular speech.

Table 9: Resynthesis performance on different ablation conditions.

| Input | FLEURS-R (20 languages) | | | | | GTSinger (OOD) | | | | | |
|---|---|---|---|---|---|---|---|---|---|---|---|
| | WER↓ | STOI↑ | PESQ↑ | UTMOS↑ | SSIM↑ | F0-PCC(r)↑ | F0-$R^2$↑ | PESQ↑ | STOI↑ | UTMOS↑ | SSIM↑ |
| Sylber | 12.343 | 0.822 | 1.287 | 2.793 | 0.928 | 0.835 | 0.355 | 0.556 | 1.200 | 2.097 | 0.890 |
| Sylber 2.0 | **6.875** | **0.929** | **2.493** | **2.748** | **0.979** | **0.936** | **0.414** | **0.687** | **1.907** | **2.122** | **0.948** |
| - w/o wSegPE | 6.982 | 0.926 | 2.485 | 2.741 | 0.977 | 0.928 | 0.389 | 0.675 | 1.841 | 2.053 | 0.945 |
| - w/o Acoustic | 10.114 | 0.765 | 1.165 | 2.570 | 0.819 | 0.810 | -2.558 | 0.510 | 1.085 | 2.120 | 0.761 |

As shown in Table 9, the original Sylber embeddings show degraded results in all metrics. This suggests that the original Sylber produces incomplete embeddings or segments that are not universally compatible. While ablating wSegPE has minimal influence in the in-domain test, it shows degraded performance in OOD settings, especially in a lower $F0 - R^2$ and UTMOS. This UTMOS is even lower than the one without acoustic encoder, thus informing positions within the segments by wSegPE is crucial for reconstructing high quality prosody. Dropping acoustic encoder yields drops in performance in all aspects, especially $F0 - R^2$ in OOD setting, having the pitch level is entirely off. However, the intelligibility score is still higher than the original Sylber with acoustic encoder. These results imply a disentanglement of contents and acoustic characteristics in Sylber 2.0.

## B.4 EMBEDDING VISUALIZATION

We select the top 50 English syllables in LibriSpeech that have most overlapping intervals with Sylber 2.0 segments. For each syllable, we extract Sylber 2.0 content and acoustic features for 1k samples found from the dataset. Then, we separately fit tSNE on content and acoustic embedding. Figure 5 shows content features are phonetically organized in the embedding showing clusters being located by phonetic properties. However, none of phonetic pattern of syllables is aligned with tSNE

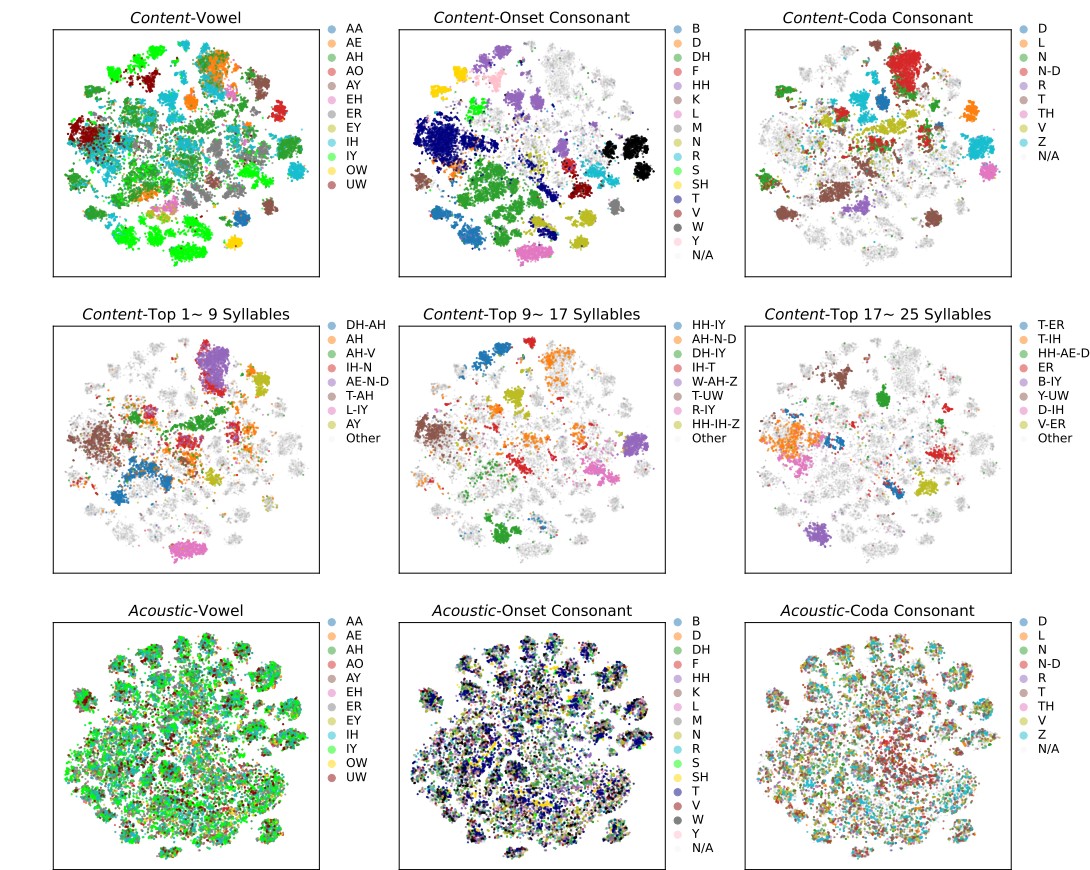

Figure 5: Visualization of Sylber 2.0 embeddings through tSNE. The top 50 syllables that are most corresponding to Sylber 2.0 are chosen and for each syllable 1K samples are retrieved from LibriSpeech. The top two rows use the content feature, with different colorization schemes for phonetic categories. The bottom uses the acoustic feature. While the manifold of the content features are phonetically structured, the acoustic feature shows no such structure.

layout of acoustic features, suggesting the successful disentanglement of the intelligible contents and acoustic information. This qualitatively shows that Sylber 2.0 learns phonetically meaningful speech embeddings, which corroborates with the results in downstream tasks.

## B.5 INDIVIDUAL RESYNTHESIS RESULTS FOR LANGUAGES IN FLEURS-R

The scores for resynthesis are denoted for each language in Table 10 and 11. Also, we denote the token frequency of all 102 languages in Table 12.

Table 10: Resynthesis performance on different languages in FLEURS-R (Part A). ↓ and ↑ represent that lower or higher values are better.

| | ca | | | | | cmn | | | | |
|---|---|---|---|---|---|---|---|---|---|---|
| Model | WER↓ | STOI↑ | PESQ↑ | UTMOS↑ | SSIM↑ | WER↓ | PESQ↑ | STOI↑ | UTMOS↑ | SSIM↑ |
| DAC | 5.27 | 1.00 | 4.49 | 3.31 | 1.00 | 6.93 | 1.00 | 4.53 | 3.16 | 1.00 |
| FACodec | 5.94 | 0.96 | 3.16 | 3.39 | 0.98 | 8.65 | 0.95 | 3.02 | 3.19 | 0.98 |
| SpeechTokenizer | 6.34 | 0.94 | 2.69 | 2.98 | 0.97 | 8.48 | 0.93 | 2.80 | 2.91 | 0.97 |
| WavTokenizer | 34.20 | 0.87 | 1.72 | 2.60 | 0.90 | 38.97 | 0.86 | 1.80 | 2.70 | 0.89 |
| Mimi | 5.45 | 0.98 | 3.70 | 3.16 | 0.98 | 7.39 | 0.97 | 3.67 | 3.00 | 0.98 |
| Sylber | 25.73 | 0.77 | 1.15 | 3.86 | 0.84 | 38.10 | 0.75 | 1.21 | 3.62 | 0.83 |
| Sylber 2.0 (Ours) | 6.37 | 0.93 | 2.35 | 2.94 | 0.97 | 8.21 | 0.91 | 2.29 | 2.96 | 0.97 |

| | de | | | | | en | | | | |
|---|---|---|---|---|---|---|---|---|---|---|
| Model | WER↓ | STOI↑ | PESQ↑ | UTMOS↑ | SSIM↑ | WER↓ | PESQ↑ | STOI↑ | UTMOS↑ | SSIM↑ |
| DAC | 4.25 | 1.00 | 4.55 | 3.29 | 1.00 | 5.45 | 1.00 | 4.54 | 4.04 | 1.00 |
| FACodec | 5.20 | 0.95 | 3.09 | 3.35 | 0.98 | 6.38 | 0.94 | 2.90 | 4.11 | 0.98 |
| SpeechTokenizer | 5.53 | 0.93 | 2.88 | 3.15 | 0.98 | 10.13 | 0.92 | 2.85 | 3.89 | 0.98 |
| WavTokenizer | 26.06 | 0.87 | 1.88 | 2.96 | 0.91 | 15.47 | 0.86 | 1.87 | 3.75 | 0.92 |
| Mimi | 4.47 | 0.97 | 3.76 | 3.18 | 0.99 | 6.14 | 0.96 | 3.72 | 3.95 | 0.98 |
| Sylber | 19.21 | 0.78 | 1.21 | 3.58 | 0.83 | 10.84 | 0.77 | 1.22 | 3.91 | 0.86 |
| Sylber 2.0 (Ours) | 5.34 | 0.92 | 2.29 | 3.21 | 0.98 | 6.46 | 0.90 | 2.20 | 3.94 | 0.97 |

| | es | | | | | fi | | | | |
|---|---|---|---|---|---|---|---|---|---|---|
| Model | WER↓ | STOI↑ | PESQ↑ | UTMOS↑ | SSIM↑ | WER↓ | PESQ↑ | STOI↑ | UTMOS↑ | SSIM↑ |
| DAC | 2.91 | 1.00 | 4.53 | 3.29 | 1.00 | 9.28 | 1.00 | 4.51 | 3.02 | 1.00 |
| FACodec | 3.21 | 0.96 | 3.34 | 3.32 | 0.98 | 11.38 | 0.95 | 3.00 | 3.09 | 0.98 |
| SpeechTokenizer | 3.13 | 0.94 | 3.06 | 2.96 | 0.97 | 12.71 | 0.92 | 2.66 | 2.75 | 0.97 |
| WavTokenizer | 14.57 | 0.87 | 1.98 | 2.73 | 0.92 | 49.28 | 0.84 | 1.75 | 2.39 | 0.91 |
| Mimi | 2.93 | 0.98 | 3.91 | 3.14 | 0.98 | 9.93 | 0.97 | 3.57 | 2.89 | 0.98 |
| Sylber | 10.66 | 0.77 | 1.18 | 3.37 | 0.84 | 32.73 | 0.73 | 1.18 | 3.30 | 0.81 |
| Sylber 2.0 (Ours) | 3.18 | 0.93 | 2.56 | 2.91 | 0.98 | 11.80 | 0.91 | 2.31 | 2.82 | 0.98 |

| | fr | | | | | id | | | | |
|---|---|---|---|---|---|---|---|---|---|---|
| Model | WER↓ | STOI↑ | PESQ↑ | UTMOS↑ | SSIM↑ | WER↓ | PESQ↑ | STOI↑ | UTMOS↑ | SSIM↑ |
| DAC | 6.34 | 1.00 | 4.52 | 3.08 | 1.00 | 8.32 | 1.00 | 4.54 | 3.01 | 1.00 |
| FACodec | 8.90 | 0.95 | 2.90 | 3.21 | 0.98 | 9.90 | 0.94 | 2.91 | 3.08 | 0.98 |
| SpeechTokenizer | 8.71 | 0.92 | 2.72 | 2.91 | 0.97 | 10.11 | 0.93 | 2.78 | 2.86 | 0.98 |
| WavTokenizer | 53.89 | 0.85 | 1.81 | 2.89 | 0.90 | 32.91 | 0.85 | 1.75 | 2.52 | 0.90 |
| Mimi | 6.66 | 0.96 | 3.64 | 2.98 | 0.98 | 8.75 | 0.97 | 3.75 | 2.90 | 0.99 |
| Sylber | 59.03 | 0.74 | 1.22 | 3.61 | 0.82 | 33.80 | 0.74 | 1.16 | 3.38 | 0.82 |
| Sylber 2.0 (Ours) | 8.92 | 0.91 | 2.28 | 2.99 | 0.97 | 10.77 | 0.92 | 2.33 | 2.88 | 0.98 |

| | it | | | | | ja | | | | |
|---|---|---|---|---|---|---|---|---|---|---|
| Model | WER↓ | STOI↑ | PESQ↑ | UTMOS↑ | SSIM↑ | WER↓ | PESQ↑ | STOI↑ | UTMOS↑ | SSIM↑ |
| DAC | 2.33 | 1.00 | 4.50 | 3.56 | 1.00 | 4.81 | 1.00 | 4.54 | 3.51 | 1.00 |
| FACodec | 2.71 | 0.96 | 3.26 | 3.59 | 0.99 | 5.44 | 0.96 | 3.25 | 3.61 | 0.98 |
| SpeechTokenizer | 2.84 | 0.94 | 2.90 | 3.18 | 0.97 | 5.70 | 0.94 | 3.00 | 3.32 | 0.97 |
| WavTokenizer | 12.83 | 0.87 | 1.99 | 2.88 | 0.92 | 21.73 | 0.89 | 1.98 | 2.97 | 0.91 |
| Mimi | 2.38 | 0.97 | 3.78 | 3.39 | 0.98 | 4.97 | 0.98 | 3.86 | 3.42 | 0.98 |
| Sylber | 7.77 | 0.76 | 1.17 | 3.63 | 0.81 | 15.54 | 0.80 | 1.18 | 3.45 | 0.83 |
| Sylber 2.0 (Ours) | 2.92 | 0.92 | 2.57 | 3.32 | 0.98 | 5.64 | 0.94 | 2.61 | 3.16 | 0.98 |

Table 11: Resynthesis performance on different languages in FLEURS-R (Part B). ↓ and ↑ represent that lower or higher values are better.

| | ko | | | | | ms | | | | |
|---|---|---|---|---|---|---|---|---|---|---|
| Model | WER↓ | STOI↑ | PESQ↑ | UTMOS↑ | SSIM↑ | WER↓ | PESQ↑ | STOI↑ | UTMOS↑ | SSIM↑ |
| DAC | 4.30 | 1.00 | 4.52 | 3.51 | 1.00 | 9.15 | 1.00 | 4.51 | 2.96 | 1.00 |
| FACodec | 5.36 | 0.96 | 3.24 | 3.63 | 0.98 | 11.07 | 0.95 | 2.69 | 3.00 | 0.98 |
| SpeechTokenizer | 4.96 | 0.94 | 2.96 | 3.25 | 0.97 | 11.85 | 0.92 | 2.50 | 2.76 | 0.98 |
| WavTokenizer | 24.42 | 0.88 | 1.97 | 2.97 | 0.90 | 48.39 | 0.84 | 1.54 | 2.39 | 0.90 |
| Mimi | 4.69 | 0.98 | 3.78 | 3.38 | 0.98 | 9.76 | 0.97 | 3.50 | 2.87 | 0.98 |
| Sylber | 17.96 | 0.80 | 1.22 | 3.61 | 0.82 | 36.17 | 0.73 | 1.13 | 3.37 | 0.81 |
| Sylber 2.0 (Ours) | 5.05 | 0.94 | 2.60 | 3.30 | 0.97 | 11.38 | 0.91 | 2.04 | 2.78 | 0.98 |

| | nb | | | | | nl | | | | |
|---|---|---|---|---|---|---|---|---|---|---|
| Model | WER↓ | STOI↑ | PESQ↑ | UTMOS↑ | SSIM↑ | WER↓ | PESQ↑ | STOI↑ | UTMOS↑ | SSIM↑ |
| DAC | 9.46 | 1.00 | 4.50 | 3.53 | 1.00 | 6.01 | 1.00 | 4.54 | 3.31 | 1.00 |
| FACodec | 10.38 | 0.96 | 3.18 | 3.62 | 0.98 | 7.54 | 0.95 | 3.03 | 3.41 | 0.97 |
| SpeechTokenizer | 10.51 | 0.94 | 2.81 | 3.14 | 0.97 | 8.28 | 0.93 | 2.77 | 3.15 | 0.97 |
| WavTokenizer | 34.07 | 0.89 | 1.92 | 2.81 | 0.92 | 49.84 | 0.86 | 1.72 | 2.95 | 0.88 |
| Mimi | 9.87 | 0.98 | 3.69 | 3.37 | 0.97 | 6.31 | 0.97 | 3.67 | 3.22 | 0.98 |
| Sylber | 20.98 | 0.80 | 1.23 | 3.58 | 0.81 | 40.40 | 0.77 | 1.15 | 3.70 | 0.81 |
| Sylber 2.0 (Ours) | 10.87 | 0.94 | 2.59 | 3.26 | 0.98 | 8.21 | 0.92 | 2.19 | 3.21 | 0.97 |

| | pl | | | | | pt | | | | |
|---|---|---|---|---|---|---|---|---|---|---|
| Model | WER↓ | STOI↑ | PESQ↑ | UTMOS↑ | SSIM↑ | WER↓ | PESQ↑ | STOI↑ | UTMOS↑ | SSIM↑ |
| DAC | 5.97 | 1.00 | 4.51 | 3.02 | 1.00 | 3.93 | 1.00 | 4.53 | 3.49 | 1.00 |
| FACodec | 7.51 | 0.95 | 2.91 | 3.07 | 0.98 | 4.28 | 0.95 | 3.08 | 3.53 | 0.98 |
| SpeechTokenizer | 8.42 | 0.93 | 2.61 | 2.85 | 0.97 | 4.43 | 0.93 | 2.96 | 3.32 | 0.98 |
| WavTokenizer | 55.72 | 0.85 | 1.66 | 2.69 | 0.90 | 13.32 | 0.87 | 1.83 | 2.95 | 0.90 |
| Mimi | 6.31 | 0.97 | 3.60 | 2.93 | 0.98 | 3.97 | 0.97 | 3.85 | 3.37 | 0.98 |
| Sylber | 55.33 | 0.77 | 1.15 | 3.47 | 0.80 | 15.23 | 0.77 | 1.18 | 3.67 | 0.84 |
| Sylber 2.0 (Ours) | 8.45 | 0.91 | 2.20 | 3.01 | 0.97 | 4.43 | 0.92 | 2.32 | 3.25 | 0.98 |

| | ru | | | | | sv | | | | |
|---|---|---|---|---|---|---|---|---|---|---|
| Model | WER↓ | STOI↑ | PESQ↑ | UTMOS↑ | SSIM↑ | WER↓ | PESQ↑ | STOI↑ | UTMOS↑ | SSIM↑ |
| DAC | 5.26 | 1.00 | 4.51 | 3.32 | 1.00 | 8.56 | 1.00 | 4.54 | 3.49 | 1.00 |
| FACodec | 5.88 | 0.95 | 2.86 | 3.34 | 0.98 | 10.44 | 0.96 | 3.15 | 3.55 | 0.98 |
| SpeechTokenizer | 6.02 | 0.93 | 2.63 | 3.04 | 0.98 | 10.31 | 0.94 | 2.93 | 3.32 | 0.97 |
| WavTokenizer | 27.58 | 0.85 | 1.72 | 2.73 | 0.90 | 44.40 | 0.87 | 1.92 | 3.05 | 0.90 |
| Mimi | 5.45 | 0.97 | 3.56 | 3.15 | 0.98 | 9.08 | 0.97 | 3.81 | 3.34 | 0.98 |
| Sylber | 24.13 | 0.75 | 1.19 | 3.56 | 0.80 | 29.91 | 0.78 | 1.16 | 3.54 | 0.83 |
| Sylber 2.0 (Ours) | 6.57 | 0.91 | 2.23 | 3.16 | 0.98 | 10.90 | 0.92 | 2.38 | 3.31 | 0.98 |

| | tr | | | | | uk | | | | |
|---|---|---|---|---|---|---|---|---|---|---|
| Model | WER↓ | STOI↑ | PESQ↑ | UTMOS↑ | SSIM↑ | WER↓ | PESQ↑ | STOI↑ | UTMOS↑ | SSIM↑ |
| DAC | 6.64 | 1.00 | 4.54 | 3.40 | 1.00 | 7.33 | 1.00 | 4.54 | 3.28 | 1.00 |
| FACodec | 8.17 | 0.95 | 3.20 | 3.40 | 0.98 | 9.61 | 0.95 | 3.00 | 3.33 | 0.98 |
| SpeechTokenizer | 7.94 | 0.93 | 2.99 | 3.19 | 0.97 | 9.72 | 0.93 | 2.83 | 3.13 | 0.97 |
| WavTokenizer | 36.87 | 0.86 | 1.90 | 2.86 | 0.89 | 44.76 | 0.86 | 1.82 | 2.97 | 0.92 |
| Mimi | 6.99 | 0.97 | 3.87 | 3.31 | 0.98 | 7.80 | 0.97 | 3.76 | 3.16 | 0.98 |
| Sylber | 27.80 | 0.76 | 1.20 | 3.53 | 0.82 | 44.59 | 0.77 | 1.16 | 3.59 | 0.81 |
| Sylber 2.0 (Ours) | 8.23 | 0.92 | 2.44 | 3.08 | 0.98 | 9.93 | 0.92 | 2.29 | 3.18 | 0.98 |

Table 12: Token frequency of individual languages in FLEURS-R (total 102 languages). Other text BPE tokenization from multilingual models are also denoted.

| Language | Sylber 2.0 | XLM-R | MT5 | UMT5 | LOLA | Whisper | Language | Sylber 2.0 | XLM-R | MT5 | UMT5 | LOLA | Whisper |
|---|---|---|---|---|---|---|---|---|---|---|---|---|---|
| af | 4.62 | 2.83 | 3.10 | 2.97 | 3.15 | 3.67 | am | 5.16 | 3.39 | 4.88 | 6.82 | 17.32 | 17.51 |
| ar | 5.71 | 3.09 | 3.93 | 2.87 | 3.48 | 5.49 | as | 5.07 | 4.56 | 5.12 | 5.61 | 8.87 | 20.28 |
| ast | 5.69 | 4.19 | 4.64 | 4.20 | 4.28 | 4.71 | az | 4.75 | 2.76 | 3.57 | 3.33 | 5.31 | 5.27 |
| be | 4.41 | 2.99 | 3.57 | 3.68 | 5.19 | 5.27 | bg | 5.31 | 3.48 | 4.24 | 3.62 | 4.34 | 5.76 |
| bn | 4.90 | 3.10 | 3.89 | 3.60 | 7.14 | 18.67 | bs | 4.35 | 2.77 | 3.61 | 3.43 | 4.16 | 4.39 |
| ca | 4.91 | 3.22 | 3.82 | 3.27 | 3.76 | 3.76 | ceb | 3.91 | 3.05 | 3.14 | 3.10 | 3.44 | 3.71 |
| ckb | 4.79 | 5.91 | 4.95 | 6.65 | 9.03 | 10.39 | cmn | 4.64 | 2.48 | 2.58 | 3.11 | 2.91 | 4.40 |
| cs | 4.50 | 2.88 | 3.47 | 2.63 | 4.59 | 4.71 | cy | 4.11 | 2.94 | 3.83 | 3.61 | 4.06 | 3.99 |
| da | 4.62 | 2.95 | 3.40 | 2.88 | 3.48 | 4.04 | de | 4.65 | 2.75 | 3.07 | 2.59 | 2.67 | 3.21 |
| el | 5.66 | 4.10 | 5.11 | 3.96 | 5.14 | 7.70 | en | 4.43 | 2.65 | 2.93 | 2.56 | 2.49 | 2.44 |
| es | 5.05 | 2.93 | 3.53 | 2.82 | 2.93 | 3.37 | et | 4.93 | 2.78 | 3.06 | 2.95 | 3.93 | 4.22 |
| fa | 4.30 | 2.13 | 2.88 | 2.13 | 2.73 | 5.13 | ff | 4.35 | 3.19 | 3.19 | 3.22 | 3.52 | 3.61 |
| fi | 4.74 | 2.68 | 2.99 | 2.63 | 3.18 | 3.95 | fil | 3.66 | 2.46 | 2.77 | 2.69 | 2.94 | 3.26 |
| fr | 5.00 | 3.83 | 4.54 | 3.64 | 3.72 | 4.29 | ga | 3.80 | 3.04 | 3.74 | 3.75 | 4.16 | 4.27 |
| gl | 5.67 | 3.42 | 4.42 | 3.59 | 3.80 | 4.27 | gu | 6.07 | 4.13 | 5.51 | 7.58 | 31.26 | 30.56 |
| ha | 3.72 | 2.33 | 2.51 | 2.64 | 2.93 | 3.04 | he | 5.35 | 3.60 | 4.28 | 3.72 | 3.98 | 6.01 |
| hi | 5.38 | 3.14 | 4.45 | 3.57 | 7.41 | 11.35 | hr | 4.94 | 3.22 | 4.20 | 3.98 | 4.81 | 5.09 |
| hu | 4.99 | 2.93 | 3.44 | 2.82 | 3.41 | 4.85 | hy | 4.77 | 3.65 | 4.57 | 5.41 | 20.16 | 14.53 |
| id | 5.26 | 2.27 | 2.87 | 2.65 | 2.74 | 3.50 | ig | 3.70 | 3.99 | 3.70 | 3.84 | 4.07 | 4.27 |
| is | 4.68 | 2.93 | 3.45 | 3.44 | 4.44 | 4.67 | it | 4.24 | 2.46 | 3.05 | 2.35 | 2.49 | 3.08 |
| ja | 4.84 | 2.49 | 2.22 | 2.45 | 2.53 | 3.87 | jv | 4.91 | 2.49 | 2.89 | 2.96 | 3.15 | 3.46 |
| ka | 5.55 | 3.67 | 4.65 | 5.47 | 5.35 | 31.69 | kam | 3.74 | 3.11 | 3.24 | 3.23 | 3.38 | 3.67 |
| kea | 4.04 | 2.96 | 3.10 | 2.95 | 3.11 | 3.41 | kk | 4.16 | 2.16 | 2.49 | 2.54 | 5.28 | 5.34 |
| km | 4.47 | 3.15 | 3.07 | 6.81 | 19.85 | 19.79 | kn | 5.19 | 3.01 | 3.47 | 5.19 | 20.51 | 16.50 |
| ko | 4.74 | 2.67 | 3.25 | 3.11 | 3.17 | 3.42 | ky | 5.43 | 3.07 | 3.82 | 4.19 | 6.41 | 7.10 |
| lb | 5.02 | 4.47 | 4.40 | 4.37 | 4.51 | 4.89 | lg | 3.69 | 2.77 | 2.77 | 2.87 | 3.06 | 3.31 |
| ln | 3.36 | 2.24 | 2.22 | 2.24 | 2.40 | 2.54 | lo | 5.31 | 3.47 | 3.49 | 9.35 | 21.15 | 28.72 |
| lt | 4.98 | 3.26 | 3.75 | 3.35 | 5.10 | 5.20 | luo | 4.60 | 3.27 | 3.36 | 3.31 | 3.49 | 3.67 |
| lv | 4.56 | 3.06 | 3.56 | 3.32 | 4.92 | 5.14 | mi | 3.65 | 2.81 | 2.82 | 2.79 | 3.03 | 3.14 |
| mk | 4.85 | 2.94 | 3.57 | 3.41 | 4.50 | 5.20 | ml | 4.96 | 2.88 | 3.11 | 4.62 | 8.15 | 21.82 |
| mn | 5.10 | 3.37 | 4.53 | 4.64 | 7.67 | 8.47 | mr | 5.10 | 2.76 | 3.80 | 3.54 | 7.41 | 11.02 |
| ms | 5.57 | 2.58 | 3.30 | 3.05 | 3.13 | 4.04 | mt | 4.41 | 4.51 | 4.31 | 4.19 | 5.01 | 5.29 |
| my | 4.18 | 3.29 | 3.29 | 5.06 | 7.14 | 28.51 | nb | 4.20 | 2.56 | 2.95 | 2.51 | 3.16 | 3.50 |
| ne | 5.60 | 2.97 | 4.25 | 4.08 | 8.45 | 12.31 | nl | 5.86 | 3.61 | 4.09 | 3.42 | 3.78 | 4.62 |
| nso | 3.73 | 2.86 | 2.83 | 2.97 | 3.13 | 3.24 | ny | 3.84 | 2.84 | 2.66 | 2.86 | 3.22 | 3.50 |
| oc | 3.97 | 2.88 | 3.15 | 2.90 | 3.04 | 3.18 | om | 4.53 | 3.47 | 3.64 | 3.61 | 3.92 | 4.16 |
| or | 5.63 | 3.70 | 8.80 | 11.28 | 30.09 | 29.93 | pa | 5.17 | 3.82 | 5.65 | 7.05 | 20.58 | 20.52 |
| pl | 5.46 | 3.63 | 4.38 | 3.34 | 3.96 | 4.85 | ps | 5.61 | 3.37 | 4.42 | 4.49 | 5.84 | 7.55 |
| pt | 4.47 | 2.68 | 3.41 | 2.57 | 2.78 | 3.11 | ro | 5.49 | 3.64 | 4.39 | 3.41 | 5.27 | 5.14 |
| ru | 4.97 | 3.03 | 3.61 | 2.95 | 3.23 | 4.08 | sd | 5.30 | 3.24 | 4.86 | 4.86 | 6.06 | 7.75 |
| sk | 4.57 | 2.97 | 3.60 | 2.97 | 4.74 | 4.78 | sl | 5.43 | 3.46 | 4.04 | 3.72 | 5.05 | 5.28 |
| sn | 3.81 | 2.91 | 2.64 | 2.95 | 3.24 | 3.43 | so | 4.83 | 3.14 | 3.68 | 3.80 | 4.48 | 4.56 |
| sr | 4.61 | 3.13 | 3.82 | 3.54 | 5.32 | 5.81 | sv | 4.94 | 2.94 | 3.38 | 2.81 | 3.25 | 3.79 |
| sw | 4.40 | 2.40 | 2.85 | 2.97 | 3.35 | 3.71 | ta | 5.64 | 3.22 | 3.32 | 3.87 | 9.19 | 10.37 |
| te | 5.73 | 3.52 | 4.20 | 6.08 | 8.97 | 21.28 | tg | 4.42 | 4.43 | 3.74 | 3.90 | 5.42 | 5.84 |
| th | 5.57 | 2.76 | 2.78 | 3.54 | 6.03 | 8.82 | tr | 4.72 | 2.45 | 2.91 | 2.32 | 3.17 | 3.54 |
| uk | 5.27 | 3.40 | 4.15 | 3.31 | 5.77 | 5.33 | umb | 3.15 | 2.10 | 2.14 | 2.16 | 2.30 | 2.48 |
| ur | 6.42 | 3.88 | 5.26 | 4.70 | 5.30 | 8.00 | uz | 5.25 | 3.40 | 3.88 | 4.10 | 4.80 | 5.03 |
| vi | 4.71 | 2.84 | 5.17 | 2.83 | 3.03 | 4.24 | wo | 3.41 | 2.56 | 2.56 | 2.57 | 2.75 | 2.83 |
| xh | 4.74 | 3.43 | 3.43 | 3.85 | 4.16 | 4.44 | yo | 3.62 | 4.14 | 4.15 | 4.12 | 4.31 | 4.76 |
| yue | 5.12 | 2.46 | 2.76 | 3.13 | 2.75 | 3.91 | zu | 4.14 | 2.92 | 2.90 | 3.26 | 3.59 | 3.81 |

