# OpenReview forum: "Sylber 2.0: A Universal Syllable Embedding"
_ICLR.cc/2026/Conference — Submitted to ICLR 2026_

### Official Review · Reviewer_3zrY · 2025-10-17

**Soundness:** 3
**Presentation:** 3
**Contribution:** 3
**Rating:** 6
**Confidence:** 3

**Summary:**

The paper introduces Sylber 2.0, a universal framework designed to encode speech at the syllable level across multiple languages. The goal is to create speech tokens that are both efficient (low temporal resolution) and universal. The framework consists of a content encoder for linguistic information, an acoustic encoder for vocal details, and a boundary detector to identify syllable breaks. A lightweight vocoder then reconstructs high-fidelity audio from these compressed embeddings. The model is trained on diverse multilingual data without using any text.

**Strengths:**

Originality: The paper is original in that it attempts to create a universal framework that learns syllabic structure directly from audio across 102 languages without any textual supervision. This moves significantly beyond previous models that were often constrained to English. The proposed architecture is also novel
Quality: The paper performed comprehensive evaluations on both reconstruction and a downstream TTS task. The authors also provided visualization to show that the model has learned consistent, syllable-like segments across different languages.
Clarity: The writing is easy to follow and the visual aids were provided.
Significance: The paper makes significant contribution in achieving a compression of speech at an average of 4.8 Hz, reducing the computational cost and memory requirements for training and running large-scale spoken language models, making them more scalable and efficient. The fact that the model works across 102 languages is also rather significant as it could serve as a powerful foundation for future multilingual speech understanding and generation systems, especially in low-resource languages.

**Weaknesses:**

1. The assessment on downstream task is rather lacking. The authors showed the performance on one TTS dataset where the proposed method's performance is not that competitive on WER. More tasks and datasets should be included to provide a better sense of the method's quality.
2. The syllable detection performance is not that high, especially the precision. Suggesting the model might be over-segmenting the speech?
3. Ablation studies on the different architecture improvement over Sylber would be nice

**Questions:**

1. Is the model learning linguistically meaningful syllables?
2. Could the authors perform more evaluations on different downstream tasks? Especially for different languages to showcase the multilinguality?

---

> ### Author Response · Authors · 2025-11-24
> **Response to W1+Q2: Lack of downstream tasks (1/2)**
>
> We greatly appreciate the reviewer’s time and effort for reviewing our paper. We have addressed the raised questions and concerns with additional experiments. Please check the response below.
>
> ### **1. Response to W1+Q2: Lack of downstream tasks**
>
>  We deeply agree that our initial scope of evaluation was narrow. To address this concern, we evaluated our model on diverse downstream tasks which include low-resource ASR, SUPERB benchmark. Furthermore we evaluate the proposed TTS with additional ablations and test corpus to further substantiate our contribution.
>
> First of all, we trained a small ASR model in low-resource settings to demonstrate multilingual capacity of Sylber 2.0 and test whether the segments contain linguistic contents. We tested 4 different languages–English, Korean, Bemba, and Quecha. While English and Korean are not low-resource languages, we chose them as English is used for the original Sylber, and Korean is chosen as it is a syllabic language, thus being readily syllable could be beneficial. Bemba and Quecha are true  low resource languages which are spoken by only 10 and 8 million people, respectively. The training data are constrained to be small: 100 hrs (English), 52 hrs (Korean), 20 hrs (Bemba), and 48 hrs (Quecha). We use RNN-T architecture for ASR since it allows shorter input length than the target, which is the case of the short token length of Sylber 2.0.
>
> | Model                     | en CER ↓ | en WER ↓ | ko CER ↓ | ko WER ↓ | bem CER ↓ | bem WER ↓ | que CER ↓ | que WER ↓ |
> |---------------------------|----------|----------|----------|----------|-----------|-----------|-----------|-----------|
> | Mel                       | 8.7     | 20.6     | 10.6     | 16.0     | 19.4      | 61.3      | 31.4      | 66.8      |
> | DAC | 36.2 | 59.3 | 100.3 | 125.3 | 30.4 | 79.5 | 51.1 | 87.3 |
> | Mimi                      | 17.1     | 32.8     | 20.9     | 31.6     | 29.6      | 80.1      | 39.9      | 76.8      |
> | Sylber                    | **7.8**  | **16.2** | 22.0     | 28.5     | 23.6      | 67.4      | 44.2      | 83.6      |
> | Sylber 2.0                | 11.7     | 24.2     | **7.2**  | **9.4**  | **12.1**  | **47.4**  | **30.1**  | **66.4**  |
> | Sylber 2.0 *Content only*      | 12.1     | 24.7     | 7.3      | 9.6      | 12.7      | 48.5      | 32.1      | 69.8      |
> | Sylber 2.0 *Acoustic only* | 31.5 | 52.9 | 22.3 | 32.4 | 39.5 | 86.6 | 48.8 | 86.1 |
>
> As shown in the above table, Sylber 2.0 outperforms the baseline speech representations other than English. For English, the original Sylber shows the best performance. However, this may be induced by the fact that Sylber is primarily trained in English and tested in-domain. In fact, the performance severely drops in other languages. Interestingly, Korean shows the lowest error rates which may be driven by the syllabic nature of the language, where the gap between Sylber 2.0 and text is narrower than others. Moreover, there is minimal performance drop when only the content part is used, suggesting that the content part is the major source of linguistic information.
>
> To provide a more comprehensive view of information carried by Sylber 2.0, we evaluated the diverse speech tasks in the SUPERB benchmark.
>
> | Model       | PR PER ↓ | KS Acc ↑ | IC Acc ↑ | SID Acc ↑ | ER Acc ↑ | ASR WER ↓ | ASR (w/ LM) WER ↓ | QbE MTWV ↑ | SF F1 ↑ | SF CER ↓ | ASV EER ↓ | SD DER ↓ |
> |-------------|-----------|----------|----------|-----------|----------|-----------|---------------------|-------------|----------|-----------|------------|-----------|
> | Mimi        | 39.79     | 93.12    | 90.61    | 34.23     | 56.05    | 91.04     | 88.65               | 0.0108      | 6.72     | 98.86     | 12.17      | 13.37     |
> | Sylber      | 88.79     | **97.11**| **99.08**| 51.25     | **65.25**| **12.04** | **8.88**            | **0.0591**  | 85.66    | **29.49** | 8.75       | 15.55     |
> | Sylber 2.0  | **14.74** | 96.85    | 98.05    | **72.03** | 62.37    | 14.23     | 10.04               | 0.0471      | **85.94**| 30.09     | **5.69**   | **6.10**  |
>
> Sylber 2.0 shows similar performance as Sylber but shows significantly better performance in speaker-related tasks: SID, ASV, and SD. Being language universal is a potential source of performance degradation in other tasks as they are all in English, though the degree is marginal. In all metrics, Sylber 2.0 outperforms Mimi.

---

> ### Author Response · Authors · 2025-11-24
> **Response to W1+Q2: Lack of downstream tasks (2/2)**
>
> Lastly, we have augmented our experiments on TTS by 1) adding an additional evaluation dataset (SeedTTS En), 2) ablation study by replacing Sylber 2.0 embeddings with Mimi or Mel spectrograms, and 3) scaling training data up to 47k hours.
>
> We chose Mimi since that has the lowest frame-rate (12.5 Hz) among the baselines considered, which is the closest to ours but still being more than twice longer. Mel spectrogram is chosen to provide reference of using raw-acoustics by having no linguistic structure other than spectral decompositions. We additionally report our TTS model trained with a larger dataset, Emilia, with 100K hours of English speech. With the same model size and number of iterations.
>
> | Model                             | #Params | Training Data     | Libri WER ↓ | Libri SIM-o ↑ | Libri UTMOS ↑ | Seed WER ↓ | Seed SIM-o ↑ | Seed UTMOS ↑ |
> |-----------------------------------|---------|-------------------|-------------|----------------|----------------|-------------|----------------|----------------|
> | **Ground Truth**                  | –       | –                 | 2.47        | 0.69           | 4.09           | 2.14        | 0.73           | –              |
> | **CosyVoice**                     | 300M    | Multi-170k        | 3.59        | 0.66           | –              | 4.08        | 0.64           | –              |
> | **CosyVoice 2**                   | 500M    | Multi-170k        | 2.47        | 0.65           | **4.35**       | 2.57        | 0.65           | –              |
> | **FireRedTTS**                    | 580M    | Multi-248k        | 2.69        | 0.47           | –              | 3.82        | 0.46           | –              |
> | **MaskGCT**                       | 1048M   | Emilia-100k       | 2.72        | **0.69**       | 3.90           | 2.62        | 0.71           | –              |
> | **F5-TTS**                        | 300M    | Emilia-100k       | 2.42        | 0.66           | 3.88           | 1.83        | 0.65           | –              |
> | **DiTAR**                         | 600M    | Emilia-100k       | 2.39        | 0.67           | 4.22           | 1.69        | 0.74           | –              |
> | **SparkTTS**                      | 500M    | Multi-100k        | –           | –              | 4.35           | 1.98        | 0.58           | –              |
> | **CLEAR-Base**                    | 439M    | Libri-50k         | 2.21        | 0.59           | 4.22           | –           | –              | –              |
> | **CLEAR-Large**                   | 686M    | Libri-50k         | **1.88**    | 0.59           | 4.22           | –           | –              | –              |
> | **SylFlow (Ours)**                | 72M     | LibriTTS-0.6k     | 3.10        | 0.31           | 4.27           | 2.62        | 0.31           | 4.19           |
> |                                   |         | +Emilia-47k       | 2.35        | 0.36           | 4.33           | 1.92        | 0.35           | 4.31           |
> | *Ablation — Replacing Sylber 2.0* |         |                   |             |                |                |             |                |                |
> | ↳ *with Mel*                      | 109M    | LibriTTS-0.6k     | 5.73        | 0.18           | 3.28           | 4.48        | 0.18           | 3.38           |
> | ↳ *with Mimi*                     | 74M     | LibriTTS-0.6k     | 10.36       | 0.30           | 2.38           | 8.21        | 0.31           | 2.49           |
>
>
> As shown in the above table, replacing Sylber 2.0 with Mimi or Mel spectrogram increases WER  by a large margin. This indicates that the gain of using Sylber 2.0 is actually largely lies in intelligibility compared to the baselines. By scaling the dataset to 100k, the model could achieve low WER as 2.35 even with the same small model size. This indicates that the low performance of SylFlow is indeed induced by being minimally scaled. Yet, within the constrained resources, Sylber 2.0 provides a greater capacity in translating text to speech, suggesting that Sylber 2.0 has a narrower gap from text compared to the previous speech coding. This is also well-supported by our new experiments on low-resource ASR, where Sylber 2.0 outperforms Mimi and Mel. The overall pattern is consistently shown in the additional evaluation corpus, SeedTTS En.

---

> ### Author Response · Authors · 2025-11-24
> **Response to W2–Potential oversegmentation by Sylber 2.0**
>
> ### **2. Response to W2–Potential oversegmentation by Sylber 2.0**
>
> Sylber 2.0 indeed oversegments to some extent. However, this is systematic as it doesn’t ignore silent tokens as Sylber 1.0 does, which is often poised at the syllable boundary.  Moreover, we believe the oversegmentation is also induced by the language universality of Sylber 2.0. Learning language-agnostic syllabification rules could bias models to lean toward the finest rule of syllabification. For example, “o-range”, a two-syllable word in English is syllabified into three syllables in Spanish or Korean, i.e., “o-ran-ge” since there is no \nj\ in the latter languages. We believe this is a feature not a bug since every language has a different phonetic system and a universal model should be agnostic to such heterogeneity. Therefore, to be universal, Sylber 2.0’s syllabification may not perfectly fit with the rules for individual languages locally defined by linguists, and produce finer segments. Moreover, the difference in phonetics across languages has been a fundamental challenge in establishing a universal syllabification rule in linguistics. We claim our model can set a universal syllabification by being acoustically grounded and self-structured, rather than relying on manually defining rules.

---

> ### Author Response · Authors · 2025-11-24
> **Response to W3: Ablation on different part of architecture**
>
> ### **3. Response to W3: Ablation on different part of architecture**
>
> We thank reviewers for the suggestion on the ablation experiments, and we trained following ablation conditions for training decoder: 1) without wSegPE, 2) without Acoustic Encoder, and 3) replacing Sylber 2.0 content encoder with Sylber 1’s segments/embedding. When using Sylber 1, we also trained the same Acoustic Encoder and wSegPE for its own decoder. Due to the limited time line, we limit the training data to be only FLEURS-R. For a fair comparison, we also retrained the main full model as well. Similar to the main analysis, we evaluated the reconstruction performance on FLEURS-R and also on GTSinger to see the capability of out-of-distribution (OOD) generalization.
>
> **FLEURS-R (20 languages)**
> | Input           | WER↓    | STOI↑  | PESQ↑  | UTMOS↑ | SSIM↑  |
> |-----------------|---------|--------|--------|--------|---------------------|
> | Sylber+wSegPE+Acoustic  | 12.343     | 0.822      | 1.287       | 2.793      | 0.928      |
> | Sylber 2.0          | **6.875** | **0.929** | **2.493** | **2.748** | **0.979** |
> | - w/o wSegPE    | 6.982      | 0.926  | 2.485  | 2.741  | 0.977  | 0.928      |
> | - w/o Acoustic    | 10.114    | 0.765  | 1.165  | 2.570  | 0.819  | 0.810      |
>
> **GTSinger (OOD)**
> | Input           |  F0-PCC(r)↑ | F0-R²↑ | PESQ↑  | STOI↑  | UTMOS↑ | SSIM↑  |
> |-----------------|---------|--------|--------|--------|--------|------------|
> | Sylber+wSegPE+Acoustic |  0.835      | 0.355  | 0.556  | 1.200  | 2.097  | 0.890  |
> | Sylber 2.0      |  **0.936** | **0.414** | **0.687** | **1.907** | **2.122** | **0.948** |
> | - w/o wSegPE    |  0.928      | 0.389  | 0.675  | 1.841  | 2.053  | 0.945  |
> | - w/o Acoustic  |  0.810      | -2.558 | 0.510  | 1.085  | 2.120  | 0.761  |
>
> *(Sylber 1 vs Sylber 2)* As shown in the table, the original Sylber embeddings show degraded results in all metrics. This suggests that the original Sylber produces incomplete embeddings or segments that are not universally compatible.
>
> *(w/ vs w/o wSegPE)* While ablating wSegPE has minimal influence in the in-domain test, it shows degraded performance in OOD settings, especially in a lower F0-R^2 and UTMOS. This UTMOS is even lower than the one without acoustic encoder, thus informing positions within the segments by wSegPE is crucial for reconstructing high quality prosody.
>
> *(w/ vs w/o Acoustic Encoder)* Dropping acoustic encoder yields drops in performance in all aspects, especially F0-R^2 in OOD setting, having  the pitch level is entirely off. However, the intelligibility score is still higher than the original Sylber with acoustic encoder. These results imply a disentanglement of contents and acoustic characteristics in Sylber 2.0.

---

> ### Author Response · Authors · 2025-11-24
> **Response to Q 1: Is the model learning linguistically meaningful syllables?**
>
> ### **4. Response to Q 1: Is the model learning linguistically meaningful syllables?**
>
> We believe this is the case for Sylber 2.0. It is indirectly shown from the ASR and TTS tasks that at least it learns information that is easily mappable to text than the previous tokenization. Specifically, we believe the major source of parameter efficiency in SylFlow is being close to the actual syllables. As syllables are by definition units of speech production, such proximity could make machine speech production easy and simple. To further provide evidence, we visually inspected the embedding space through tSNE (Figure 5 in the revised manuscript), and the embedding space of content part of Sylber 2.0 is indeed phonetically organized. Altogether, we strongly believe Sylber 2.0 is learning linguistically meaningful syllables given our results and observations.

---

### Official Review · Reviewer_RWPH · 2025-10-28

**Soundness:** 3
**Presentation:** 2
**Contribution:** 3
**Rating:** 6
**Confidence:** 4

**Summary:**

This paper introduces Sylber 2.0, a framework for universal speech tokenization. The primary goal is to create a speech representation that is both highly efficient (targeting a ~5 Hz syllabic rate) and high-fidelity, overcoming the limitations of prior work like the original Sylber, which was constrained to English and lacked acoustic detail.

The core methodological novelty is a new, disentangled token structure composed of three parts: duration (d), linguistic content (C), and acoustic information (A). This is enabled by three modules that include:
- A syllable-guided acoustic encoder (a CNN + Transformer stack) that runs in parallel to the content encoder, capturing speaker and style information.
- A trained boundary detector that replaces a slower, iterative segmentation algorithm, enabling faster, parallel processing.
- A within-segment positional encoding (wSegPE), a method to help the vocoder reconstruct audio from variable-length syllable segments.

The authors claim this new framework can compress diverse, multilingual speech (across 102 languages) to an average of 4.8 Hz while enabling high-fidelity reconstruction that substantially outperforms the original Sylber and remains competitive with high-frequency tokenizers.

**Strengths:**

- The goal of creating a universal, high-fidelity, and highly compressed speech token is a critical and high-impact challenge for the spoken language modeling community. Success here would enable models to process much longer speech contexts efficiently.

- The paper introduces several intelligent and specific methodological ideas. The central concept of a disentangled (d, C, A) token is elegant. The syllable-guided acoustic encoder is a specific, new architecture designed to solve the well-known problem of acoustic information being "marginalized out" by self-supervised content encoders. Furthermore, the wSegPE is a clever technical solution to the inherent conflict between variable-length syllabic tokens and fixed-rate vocoders.

**Weaknesses:**

- The paper lacks empirical validation for its core contribution. The acoustic encoder ('A' token) is presented as the key innovation for achieving high-fidelity reconstruction. However, the paper provides no ablation studies to prove its impact. The main comparison in Table 2 is against the original "Sylber," which is not an apples-to-apples comparison. The gains are confounded by multiple variables: a new (multilingual) dataset, a new (and likely better) vocoder, and the removal of silent masking. It is difficult to determine if the novel acoustic encoder contributed significantly, or if the gains are simply from the new dataset and vocoder.

- The paper's core SSL methods, frame-wise self-distillation and self-segmentation distillation, are explicitly borrowed from prior work. The novelty lies in the new architecture (the (A) encoder) and the boundary detector (wSegPE).

**Questions:**

- To properly evaluate the impact of your individual methodological contributions, could you provide an ablation study where you isolate each architectural novelty? For instance, starting from a common baseline (e.g., Sylber 1.0 + new dataset).
- Regarding the high 3.29 WER in the TTS task (Table 3): Do you believe this is a limitation of your small SylFlow model, or does it suggest that your syllable-level token, by its very nature, creates a "fidelity ceiling" by smoothing essential, sub-syllabic phonetic details?

---

> ### Author Response · Authors · 2025-11-24
> **Response to W1+Q1: lack of ablation experiments**
>
> We highly appreciate your time and effort on reviewing our work. We acknowledge that our core contribution does not lie on the technical side, rather, our work is the first demonstration of the universal syllabic structure of spoken language that naturally emerges from any human annotation. Since the universal syllable has long been reputed in linguistics, we believe our results greatly contribute to understanding natural representation of speech.
>
> Following the suggestions raised by the reviewer, we have conducted ablation analyses to decompose technical contributions of each suggested component.
>
> ### **1. Response to W1+Q1: lack of ablation experiments**
>
>  We thank reviewers for the suggestion on the ablation experiments, and we trained following ablation conditions for training decoder: 1) without wSegPE, 2) without Acoustic Encoder, and 3) replacing Sylber 2.0 content encoder with Sylber 1’s segments/embedding. When using Sylber 1, we also trained the same Acoustic Encoder and wSegPE for its own decoder. Due to the limited time line, we limit the training data to be only FLEURS-R. For a fair comparison, we also retrained the main full model as well. Similar to the main analysis, we evaluated the reconstruction performance on FLEURS-R and also on the singing data corpus, GTSinger. This dataset is suitable to see the capability of out-of-distribution (OOD) generalization since the syllable durations in singing are significantly different from regular speech.
>
> **FLEURS-R (20 languages)**
> | Input           | WER↓    | STOI↑  | PESQ↑  | UTMOS↑ | SSIM↑  |
> |-----------------|---------|--------|--------|--------|---------------------|
> | Sylber+wSegPE+Acoustic  | 12.343     | 0.822      | 1.287       | 2.793      | 0.928      |
> | Sylber 2.0          | **6.875** | **0.929** | **2.493** | **2.748** | **0.979** |
> | - w/o wSegPE    | 6.982      | 0.926  | 2.485  | 2.741  | 0.977  | 0.928      |
> | - w/o Acoustic    | 10.114    | 0.765  | 1.165  | 2.570  | 0.819  | 0.810      |
>
> **GTSinger (OOD)**
> | Input           |  F0-PCC(r)↑ | F0-R²↑ | PESQ↑  | STOI↑  | UTMOS↑ | SSIM↑  |
> |-----------------|---------|--------|--------|--------|--------|------------|
> | Sylber+wSegPE+Acoustic |  0.835      | 0.355  | 0.556  | 1.200  | 2.097  | 0.890  |
> | Sylber 2.0      |  **0.936** | **0.414** | **0.687** | **1.907** | **2.122** | **0.948** |
> | - w/o wSegPE    |  0.928      | 0.389  | 0.675  | 1.841  | 2.053  | 0.945  |
> | - w/o Acoustic  |  0.810      | -2.558 | 0.510  | 1.085  | 2.120  | 0.761  |
>
> *(Sylber 1 vs Sylber 2)* As shown in the table, the original Sylber embeddings show degraded results in all metrics. This suggests that the original Sylber produces incomplete embeddings or segments that are not universally compatible.
>
> *(w/ vs w/o wSegPE)* While ablating wSegPE has minimal influence in the in-domain test, it shows degraded performance in OOD settings, especially in a lower F0-R^2 and UTMOS. This UTMOS is even lower than the one without acoustic encoder, thus informing positions within the segments by wSegPE is crucial for reconstructing high quality prosody.
>
> *(w/ vs w/o Acoustic Encoder)* Dropping acoustic encoder yields drops in performance in all aspects, especially F0-R^2 in OOD setting, having  the pitch level is entirely off. However, the intelligibility score is still higher than the original Sylber with acoustic encoder. These results imply a disentanglement of contents and acoustic characteristics in Sylber 2.0.
>
> These results suggest necessity of each component in Sylber 2.0 architecture.

---

> > ### Comment · Reviewer_RWPH · 2025-11-26
> >
> > The authors have addressed the lack of empirical validation by providing comprehensive ablation studies, clearly demonstrating the necessity of the Acoustic Encoder and wSegPE for preserving prosody and ensuring OOD generalization. Moreover, the additional experiments with scaled training data effectively resolve the concerns regarding a potential 'fidelity ceiling,' showing that the method is competitive with SOTA models. I have increased my score.

---

> ### Author Response · Authors · 2025-11-24
> **Response to W2: Lack of novelty**
>
> ### **2. Response to W2: Lack of novelty**
>
> As stated above, the core novelty of our work is providing a conceptual leap in understanding and modeling speech representation. Unlike dense frame tokenizations, we demonstrate that speech with any language or styles can be naturally structured in segments, providing interpretable handles of time-continuous acoustic signals and reducing token length at unprecedented levels. The previous works (Sylber and SyllableLM) have demonstrated this in English but provide significantly incomplete representations. And achieving language and style universality is non-trivial given the long-held controversy in linguistics regarding language-agnostic phonology. As shown in the ablation experiments in the previous response to W1+Q1, our technical solutions effectively provide a solution to achieve both high-fidelity reconstruction and universality.

---

> ### Author Response · Authors · 2025-11-24
> **Response to Q2: High WER in TTS (1/2)**
>
> ### **3. Response to Q2: High WER in TTS**
>
> We have conducted more experiments on TTS by 1) adding ablation and 2) increasing training data size. As a result, we found that the main source of high WER is the limited training data size. When adding more 47k hours of training data from Emilia, the WER goes down to 2.35% which closes the gap from the large SOTA models. This is consistent with an additional test set by SeedTTS, where SylFlow shows 1.92 WER. Given the previous best models show 1.88 and 1.83 on those eval sets, WERs by our model are impressively competitive on accuracy of generated text using only 72M parameters. The below table organizes the results.
>
> | Model                             | #Params | Training Data     | Libri WER ↓ | Libri SIM-o ↑ | Libri UTMOS ↑ | Seed WER ↓ | Seed SIM-o ↑ | Seed UTMOS ↑ |
> |-----------------------------------|---------|-------------------|-------------|----------------|----------------|-------------|----------------|----------------|
> | **Ground Truth**                  | –       | –                 | 2.47        | 0.69           | 4.09           | 2.14        | 0.73           | –              |
> | **CosyVoice**                     | 300M    | Multi-170k        | 3.59        | 0.66           | –              | 4.08        | 0.64           | –              |
> | **CosyVoice 2**                   | 500M    | Multi-170k        | 2.47        | 0.65           | **4.35**       | 2.57        | 0.65           | –              |
> | **FireRedTTS**                    | 580M    | Multi-248k        | 2.69        | 0.47           | –              | 3.82        | 0.46           | –              |
> | **MaskGCT**                       | 1048M   | Emilia-100k       | 2.72        | **0.69**       | 3.90           | 2.62        | 0.71           | –              |
> | **F5-TTS**                        | 300M    | Emilia-100k       | 2.42        | 0.66           | 3.88           | 1.83        | 0.65           | –              |
> | **DiTAR**                         | 600M    | Emilia-100k       | 2.39        | 0.67           | 4.22           | 1.69        | 0.74           | –              |
> | **SparkTTS**                      | 500M    | Multi-100k        | –           | –              | 4.35           | 1.98        | 0.58           | –              |
> | **CLEAR-Base**                    | 439M    | Libri-50k         | 2.21        | 0.59           | 4.22           | –           | –              | –              |
> | **CLEAR-Large**                   | 686M    | Libri-50k         | **1.88**    | 0.59           | 4.22           | –           | –              | –              |
> | **SylFlow (Ours)**                | 72M     | LibriTTS-0.6k     | 3.10        | 0.31           | 4.27           | 2.62        | 0.31           | 4.19           |
> |                                   |         | +Emilia-47k       | 2.35        | 0.36           | 4.33           | 1.92        | 0.35           | 4.31           |
> | *Ablation — Replacing Sylber 2.0* |         |                   |             |                |                |             |                |                |
> | ↳ *with Mel*                      | 109M    | LibriTTS-0.6k     | 5.73        | 0.18           | 3.28           | 4.48        | 0.18           | 3.38           |
> | ↳ *with Mimi*                     | 74M     | LibriTTS-0.6k     | 10.36       | 0.30           | 2.38           | 8.21        | 0.31           | 2.49           |
>
>
> However, the limited model size affects more on cloning prompted speaker style. we erroneously reported the SIM-o score in the original manuscript, as we mistakenly used WavLM-Base-Plus-SV whereas the baselines use WavLM-TDNN for the speaker encoder.
> The fixed results show a huge gap from the SOTA TTS models in SIM-o scores, indicating that our model is far short in cloning voice, which could be resolved by scaling the model parameters. (We have modified the main text accordingly.)
>
> For the ablation, we replaced Sylber 2.0 with Mimi and Mel spectrograms to show Sylber 2.0 provides a better, efficient representation for speech generative models.  We chose Mimi since that has the lowest frame-rate (12.5 Hz) among the baselines considered, which is the closest to ours but still being more than twice longer. Mel spectrogram is chosen to provide reference of using raw-acoustics by having no linguistic structure other than spectral decompositions.

---

> ### Author Response · Authors · 2025-11-24
> **Response to Q2: High WER in TTS (2/2)**
>
> As shown in the above table, replacing Sylber 2.0 with Mimi or Mel spectrogram increases WER  by a large margin. This indicates that the gain of using Sylber 2.0 is actually largely lies in intelligibility compared to the baselines. By scaling the dataset to 100k, the model could achieve low WER as 2.35 even with the same small model size. This indicates that the low performance of SylFlow is indeed induced by being minimally scaled. Yet, within the constrained resources, Sylber 2.0 provides a greater capacity in translating text to speech, suggesting that Sylber 2.0 has a narrower gap from text compared to the previous speech coding. This is also well-supported by our new experiments on low-resource ASR (Section 6.2 in the revised manuscript), where Sylber 2.0 outperforms Mimi and Mel.
>
> [1] Anastassiou, P., Chen, J., Chen, J., Chen, Y., Chen, Z., Chen, Z., ... & Zhuang, X. (2024). Seed-tts: A family of high-quality versatile speech generation models. arXiv preprint arXiv:2406.02430.

---

### Official Review · Reviewer_F9nD · 2025-10-31

**Soundness:** 2
**Presentation:** 3
**Contribution:** 2
**Rating:** 4
**Confidence:** 4

**Summary:**

This paper proposes Sylber 2.0, a syllable-level speech coding model, which is built upon the original Sylber. By introducing syllable-level acoustic embedding and vocoder, Sylber 2.0 enables better linguistic coverage and higher reconstruction quality than Sylber. Experiments on speech reconstruction show that it can achieve performance comparable to previous higher-frequency speech tokenizers. Zero-shot TTS experiments indicate that it can obtain result comparable to other TTS models, with much less training resources.

**Strengths:**

1) The proposed Sylber 2.0 achieves an impressive speech compression rate with token frequency of around 5Hz on many languages, while retaining both linguistic and acoustic details. Extensive experiments and analysis manifest the effectiveness of this method.
2) This paper provides detailed implementation description, and presents good academic expression, data visualization, and result analysis.

**Weaknesses:**

1) The overall novelty is limited. This work extends the original Sylber system by adding the acoustic encoder and vocoder, and also improves the training process of content encoder. These changes are incremental.
2) This paper doesn't report quantitative ablation results of the proposed changes to the original Sylber.

**Questions:**

1) In Table 3, how about the TTS results of the original Sylber?
2) This system is claimed to compress speech into syllable-level embedding. Should it demonstrate greater benefits in terms of the WER metric for resynthesis and TTS?
3) Though Table 6 reports the RTF of extracting the content embeddings, as the system looks complicated, I am also concerned about overall efficiency of this system when compared to other speech encoding/tokenizer models and also the original Sylber.

---

> ### Author Response · Authors · 2025-11-24
> **Response to W1–Limited novelty**
>
> We sincerely appreciate the reviewer’s time and effort for reviewing our paper. We have addressed the concerns and questions as below.
>
> ### **1. Response to W1–Limited novelty**
>
>  We appreciate the reviewer’s comment regarding the novelty. We would like to emphasize that our contribution is primarily conceptual rather than architectural. Specifically, our work demonstrates that a universal syllabification principle can emerge directly from the speech signal without any textual supervision. Our model learns syllable-level units that can represent speech from any language, even though these units are continuous and not quantized.
>
> However, achieving such universality is non-trivial. Linguistics has long debated—but not established—the existence of a single universal rule of syllabification. Given the high heterogeneity across languages, it was unclear whether previous demonstrations of syllabic tokenization in English would naturally extend to all languages. In fact, simply replicating Sylber in other languages did not work. To address this, we carefully designed each component of the encoding and decoding processes, even though some parts of the pipeline are similar to Sylber. The strong results in syllable boundary detection, high-fidelity reconstruction, and the visualization in Figure 4 indicate that the learned units capture distinctive linguistic structure, going beyond simple low-level acoustic cues. The new tSNE analysis in Figure 5 in the revised manuscript on the embedding corroborates this statement.
>
> Thus, while some architectural changes may appear incremental, the conceptual contribution—demonstrating universal phonetic syllable emergence from raw speech—is substantial and, to our knowledge, has not been previously achieved.

---

> ### Author Response · Authors · 2025-11-24
> **Response to W2: Lack of ablation result**
>
> ### **2. Response to W2: Lack of ablation result**
>
>  Thank you for raising concerns about the lack of ablation experiments. During the rebuttal phase, we have conducted experiments on the following ablation conditions for the reconstruction: 1) without wSegPE, 2) without Acoustic Encoder, and 3) replacing Sylber 2.0 content encoder with Sylber 1’s segments/embedding. For the original Sylber, its own acoustic encoder is trained as well using the same architecture of Sylber 2.0 acoustic encoder. We limit the resources by using only FLEURS-R for training. We trained these ablation conditions along with the full Sylber 2.0 embeddings to give a fair comparison. We additionally evaluated the reconstruction performance on singing speech (GTSinger) to see if the capability of out-of-distribution (OOD) generalization has dramatically different prosodic patterns .
>
> **FLEURS-R (20 languages)**
> | Input           | WER↓    | STOI↑  | PESQ↑  | UTMOS↑ | SSIM↑  |
> |-----------------|---------|--------|--------|--------|---------------------|
> | Sylber+wSegPE+Acoustic  | 12.343     | 0.822      | 1.287       | 2.793      | 0.928      |
> | Sylber 2.0          | **6.875** | **0.929** | **2.493** | **2.748** | **0.979** |
> | - w/o wSegPE    | 6.982      | 0.926  | 2.485  | 2.741  | 0.977  | 0.928      |
> | - w/o Acoustic    | 10.114    | 0.765  | 1.165  | 2.570  | 0.819  | 0.810      |
>
> **GTSinger (OOD)**
> | Input           |  F0-PCC(r)↑ | F0-R²↑ | PESQ↑  | STOI↑  | UTMOS↑ | SSIM↑  |
> |-----------------|---------|--------|--------|--------|--------|------------|
> | Sylber+wSegPE+Acoustic |  0.835      | 0.355  | 0.556  | 1.200  | 2.097  | 0.890  |
> | Sylber 2.0      |  **0.936** | **0.414** | **0.687** | **1.907** | **2.122** | **0.948** |
> | - w/o wSegPE    |  0.928      | 0.389  | 0.675  | 1.841  | 2.053  | 0.945  |
> | - w/o Acoustic  |  0.810      | -2.558 | 0.510  | 1.085  | 2.120  | 0.761  |
>
> (Sylber 1 vs Sylber 2.0) As shown in the table, the original Sylber embeddings show degraded results in all metrics. This suggests that the original Sylber produces incomplete embeddings or segments that are not language universal.
>
> (w/ vs w/o wSegPE) While ablating wSegPE has minimal influence in the in-domain test, it shows degraded performance in OOD settings, especially in a lower F0-R^2 and UTMOS. This UTMOS is even lower than the one without acoustic encoder, thus informing positions within the segments is crucial for reconstructing high quality prosody.
>
> (w/ vs w/o Acoustic Encoder) Dropping acoustic encoder yields drops in performance in all aspects, especially F0-R^2 in OOD setting, meaning the pitch level is entirely off. However, the intelligibility score is still higher than the original Sylber with acoustic encoder. These results imply a disentanglement of contents and acoustic characteristics in Sylber 2.0.
>
> We put more focus on the decoder side since the content encoding part is not much different from the original Sylber as noted by the reviewer. The most different procedure is the first step of using frame-wise self-distillation rather than sentence-wise self-distillation to make initial rough syllabic structure visible from (m)HuBERT. The latter method in the original Sylber was not replicable in the multilingual settings thus that we are not able to conduct ablation study regarding this. (And this was the motivation to use an alternative method.) Moreover, the boundary detector is replacing the segmentation algorithm to enable a fast batched inference, which is compared in a more detailed real-time factor (RTF) analysis in the below response to Q3. Lastly, keeping silent masking is apparently suboptimal as it masks out low gain true syllables, thus it was opted out in Sylber 2.0.

---

> ### Author Response · Authors · 2025-11-24
> **Response to Q1: TTS results of Sylber 1 (1/2)**
>
> ### **3. Response to Q1: TTS results of Sylber 1**
>
> We tried the same AR flow-based TTS architecture for the original Sylber token, but the model couldn’t generate intelligible speech. The major issue is that the original Sylber has an additional duration value that indicates silence following the main segment (also shown in the top row of Figure 4), which is not well generated, producing disconnected sounds.
>
> Instead, we conducted experiments using other representations: Mimi and Mel spectrogram.  We chose Mimi since that has the lowest frame-rate (12.5 Hz) among the baselines considered. We use the encoder and decoder of Mimi and trained the same TTS model switching the Sylber 2.0 embedding with Mimi embedding. For the Mel spectrogram, simply switching target embedding with a dense frame rate (50Hz) was not successful. Thus, we have slightly modified the architecture. In particular, we chunk every 10 frames which are weighted averaged using a convolutional layer before fed to the LM backbone. The rectified flow is then generating the whole 10 frames at once from the next token embedding from the LM. This chunking also makes the unit of LM operation at 5Hz, matching the temporal resolution of Sylber 2.0, thus providing a more fair comparison. We use a pretrained Mel vocoder by Du et al. (2024) [1]. In addition to the LibriSpeech-PC test set, we evaluated the models on the SeedTTS English test set as well.
>
> | Model                             | #Params | Training Data     | Libri WER ↓ | Libri SIM-o ↑ | Libri UTMOS ↑ | Seed WER ↓ | Seed SIM-o ↑ | Seed UTMOS ↑ |
> |-----------------------------------|---------|-------------------|-------------|----------------|----------------|-------------|----------------|----------------|
> | **Ground Truth**                  | –       | –                 | 2.47        | 0.69           | 4.09           | 2.14        | 0.73           | –              |
> | **CosyVoice**                     | 300M    | Multi-170k        | 3.59        | 0.66           | –              | 4.08        | 0.64           | –              |
> | **CosyVoice 2**                   | 500M    | Multi-170k        | 2.47        | 0.65           | **4.35**       | 2.57        | 0.65           | –              |
> | **FireRedTTS**                    | 580M    | Multi-248k        | 2.69        | 0.47           | –              | 3.82        | 0.46           | –              |
> | **MaskGCT**                       | 1048M   | Emilia-100k       | 2.72        | **0.69**       | 3.90           | 2.62        | 0.71           | –              |
> | **F5-TTS**                        | 300M    | Emilia-100k       | 2.42        | 0.66           | 3.88           | 1.83        | 0.65           | –              |
> | **DiTAR**                         | 600M    | Emilia-100k       | 2.39        | 0.67           | 4.22           | 1.69        | 0.74           | –              |
> | **SparkTTS**                      | 500M    | Multi-100k        | –           | –              | 4.35           | 1.98        | 0.58           | –              |
> | **CLEAR-Base**                    | 439M    | Libri-50k         | 2.21        | 0.59           | 4.22           | –           | –              | –              |
> | **CLEAR-Large**                   | 686M    | Libri-50k         | **1.88**    | 0.59           | 4.22           | –           | –              | –              |
> | **SylFlow (Ours)**                | 72M     | LibriTTS-0.6k     | 3.10        | 0.31           | 4.27           | 2.62        | 0.31           | 4.19           |
> |                                   |         | +Emilia-47k       | 2.35        | 0.36           | 4.33           | 1.92        | 0.35           | 4.31           |
> | *Ablation — Replacing Sylber 2.0* |         |                   |             |                |                |             |                |                |
> | ↳ *with Mel*                      | 109M    | LibriTTS-0.6k     | 5.73        | 0.18           | 3.28           | 4.48        | 0.18           | 3.38           |
> | ↳ *with Mimi*                     | 74M     | LibriTTS-0.6k     | 10.36       | 0.30           | 2.38           | 8.21        | 0.31           | 2.49           |
>
> As shown in the above table, replacing Sylber 2.0 with Mimi or Mel spectrogram degrades the performance by large margin. Interestingly, using Mel is relatively good at generating intelligible sounds while bad at cloning the reference voice, but the pattern was opposite in Mimi.
>
> We additionally report our TTS model trained with a larger dataset, Emilia, with 47k hours of English speech. With the same model size and number of iterations (1M; 2000 tokens per batch), adding a scaled dataset improves intelligibility and speaker similarity.

---

> ### Author Response · Authors · 2025-11-24
> **Response to Q1: TTS results of Sylber 1 (2/2)**
>
> Also, the original SIM-o score we reported was 0.67 which was an error that we used a different speaker embedding model, WavLM-Base-Plus-SV, that generates consistently higher than WavLM-TDNN, which is used for existing TTS models. The gap between the previous model regarding speaker similarity is huge, which we believe is due to the limited number of parameters of the model. We have constrained the model size to be minimal as our main focus is more of proof-of-concept than achieving SOTA TTS. However we believe our new findings on ablation studies indicate promising utilities of Sylber 2.0 in generative modeling. Moreover, the SylFlow still produces highly accurate and high quality speech which shows competitive WER and UTMOS compared to large SOTA models, using only 72 M parameters. Thus, we strongly believe that Sylber 2.0 enables efficient speech generative modeling. We have modified the main text to accurately convey our findings.
>
> [1] Zhihao Du, Yuxuan Wang, Qian Chen, Xian Shi, Xiang Lv, Tianyu Zhao, Zhifu Gao, Yexin Yang, Changfeng Gao, Hui Wang, et al. Cosyvoice 2: Scalable streaming speech synthesis with large language models. arXiv preprint arXiv:2412.10117, 2024b.

---

> ### Author Response · Authors · 2025-11-24
> **Response to Q3: RTF**
>
> ### **4. Response to Q3: RTF**
>
> We conducted a more detailed decomposition of RTF which is shown in the below table. In addition to Sylber, we included DAC and Mimi for dense tokenizations and mHuBERT and WavLM-Large for as reference of general speech encoding models.
>
> | Batchsize | Model        | Content ↓ | Segmentation ↓ | Acoustic ↓ | Encoding Total ↓  | Decoding RTF ↓ | E2E RTF ↓ |
> |-----------|-------------|-----------|----------------|------------|----------|----------------|------------|
> | 1         | mHuBERT     | --        | --             | --         | 0.00213  | --             | --         |
> |  1        | WavLM-Large | --        | --             | --         | 0.00404  | --             | --         |
> |  1        | DAC         | --        | --             | --         | 0.00540  | 0.00212        | 0.00752    |
> |   1        | Mimi        | --        | --             | --         | 0.00441  | 0.00202        | 0.00643    |
> |   1        | Sylber      | 0.00184   | 0.00344        | 0.01290    | 0.01818  | 0.01800        | 0.03618    |
> |   1       | Sylber 2.0  | 0.00211   | 0.00290        | 0.00269    | 0.00769  | 0.00165        | 0.00935    |
> | 32        | mHuBERT     | --        | --             | --         | 0.00027  | --             | --         |
> | 32          | WavLM-Large | --        | --             | --         | 0.00056  | --             | --         |
> | 32          | DAC         | --        | --             | --         | 0.00379  | 0.00635        | 0.01014    |
> | 32          | Mimi        | --        | --             | --         | 0.00076  | 0.00128        | 0.00204    |
> |  32      | Sylber      | 0.00027   | 0.00204        | 0.09979    | 0.10210  | 0.01089        | 0.11299    |
> |  32      | Sylber 2.0  | 0.00029   | 0.00142        | 0.00144    | 0.00315  | 0.00158        | 0.00473    |
>
>
> Compared to Sylber, Sylber 2.0 outperforms in both encoding and decoding since Sylber leverages a slow off-the-shelf speaker model and generates through multiple iterations of conditional flow-matching model.
>
> As our model has an ad hoc segmentation algorithm and non-uniform averaging/expanding mechanisms, Sylber 2.0 is slower in encoding than the straight feed-forward models, mHuBERT and WavLM-Large. However, the gap is narrower compared to Mimi or DAC, except in the batch case of DAC which is slower than Sylber 2.0. Moreover, the speed of decoding is faster or compatible with Mimi or DAC. This indicates that Sylber 2.0 can be utilized without heavy computations despite the complicated architecture.

---

### Official Review · Reviewer_QKpL · 2025-11-04

**Soundness:** 2
**Presentation:** 2
**Contribution:** 2
**Rating:** 2
**Confidence:** 4

**Summary:**

This paper presents Sylber 2.0, a universal framework for encoding speech into syllable-level embeddings at ~5 Hz frequency. The key innovation is extending syllable-based speech tokenization from English-only to 102+ languages while significantly improving reconstruction quality. The system learns syllable segmentation through self-supervised learning without text supervision, representing each syllable with three components: content embeddings (linguistic information), acoustic embeddings (speaker/style), and duration tokens. A lightweight vocoder reconstructs 24 kHz audio from these compressed representations. The authors demonstrate near-lossless compression performance comparable to high-frequency baselines (86 Hz) and showcase practical utility by training a small 72M parameter TTS model (SylFlow) that matches SOTA models 5-10× larger.

**Strengths:**

1. Achieves lowest reported token frequency (4.8 Hz average) for multilingual speech, dramatically reducing computational costs for downstream modeling compared to existing methods (12.5-86 Hz)

2.  Successfully learns syllabic structure across 102 languages without text supervision, addressing the major limitation of prior work (Sylber 1.0) which only handled English

**Weaknesses:**

1. **Insufficient downstream task validation**: The paper proposes a new speech representation primarily motivated by speech language modeling, yet only a small portion (lines 462-476) demonstrates its usage. Only TTS results are shown, which is insufficient since TTS is relatively simple and can be trained effectively with Mel-Spectrogram + Vocoder without any tokenization. The paper needs to justify the benefits of the proposed embedding for more diverse downstream tasks and identify which tasks would actually benefit from syllable-level representations

2. **Overclaimed TTS efficiency benefits**: The claim that the proposed embedding enables training smaller models with less data for TTS is overclaimed. It is well-established that good TTS models can be trained on small datasets like LJSpeech (40h) using Tacotron2 by overfitting to a single speaker, so requiring fewer parameters for smaller datasets is expected. The evaluation should be conducted on standard benchmarks like seed-tts-eval and include comparisons with recent baselines such as SparkTTS (https://github.com/SparkAudio/Spark-TTS) to show it can modeling diverse speaking style, speakers, languages, etc.

3. **Unclear advantage over model-free representations**: For multilingual speech representation, mel-spectrograms provide lossless, model-free representations. The fundamental issue with model-based compressors is generalization. The paper does not adequately justify why syllable embeddings should be preferred over mel-spectrograms for TTS, especially when an I-STFT based vocoder (e.g., https://github.com/gemelo-ai/vocos) can transform mel-spectrograms back to audio without requiring learned compression

**Questions:**

N/A

---

> ### Author Response · Authors · 2025-11-24
> **Response to Insufficient downstream task validation (1/2)**
>
> We deeply appreciate your time for reviewing and comments on our work. We have addressed the raised weaknesses with additional experiments and evaluations.
>
>
>
>
> ### 1. **Response to Insufficient downstream task validation**
>
> We acknowledge that our initial scope of downstream tasks was limited and we thank the reviewer for raising this concern. We evaluated our model low-resource ASR and diverse speech tasks in the SUPERB benchmark [1]. Due to the limited time window, we have limited the scope of baselines to be Mel spectrogram (Mel), Mimi, or the original Sylber. The comparison over Mel can provide the benefit of abstracted knowledge learned by our model over the model-free representation of speech. Mimi is selected since that produces the lowest token rate among the previous SOTA speech tokenizations. As we aim to provide tokens at an extreme rate, we believe this is the right counterpart in a dense tokenization regime.
>
> We chose low-resource ASR since this task can evaluate whether Sylber 2.0 learned linguistically useful structures other than merely copying acoustic signals. In particular, we trained a small ASR for each of 4 different languages–English, Korean, Bemba, and Quecha. While English and Korean are not low-resource languages, we chose them as English is used for the original Sylber, and Korean is chosen as it is a syllabic language. Bemba and Quecha are true low resource languages which are spoken by only 10 and 8 million people, respectively. The training data are constrained to be small: 100 hrs (English), 52 hrs (Korean), 20 hrs (Bemba), and 48 hrs (Quecha). (More details can be found in the Appendix A.3 of the revised manuscript.)
>
> | Model                     | en CER ↓ | en WER ↓ | ko CER ↓ | ko WER ↓ | bem CER ↓ | bem WER ↓ | que CER ↓ | que WER ↓ |
> |---------------------------|----------|----------|----------|----------|-----------|-----------|-----------|-----------|
> | Mel                       | 8.7     | 20.6     | 10.6     | 16.0     | 19.4      | 61.3      | 31.4      | 66.8      |
> | DAC | 36.2 | 59.3 | 100.3 | 125.3 | 30.4 | 79.5 | 51.1 | 87.3 |
> | Mimi                      | 17.1     | 32.8     | 20.9     | 31.6     | 29.6      | 80.1      | 39.9      | 76.8      |
> | Sylber                    | **7.8**  | **16.2** | 22.0     | 28.5     | 23.6      | 67.4      | 44.2      | 83.6      |
> | Sylber 2.0                | 11.7     | 24.2     | **7.2**  | **9.4**  | **12.1**  | **47.4**  | **30.1**  | **66.4**  |
> | Sylber 2.0 *Content only*      | 12.1     | 24.7     | 7.3      | 9.6      | 12.7      | 48.5      | 32.1      | 69.8      |
> | Sylber 2.0 *Acoustic only* | 31.5 | 52.9 | 22.3 | 32.4 | 39.5 | 86.6 | 48.8 | 86.1 |
>
> As shown in the above table, Sylber 2.0 outperforms the baseline speech representations other than English. (We chose Mimi and DAC for comparison since DAC is the best at resynthesis quality and Mimi is shortest in frame lengths among the previous tokens.) For English, the original Sylber shows the best performance. However, this may be induced by the fact that Sylber is primarily trained in English and tested in-domain corpus. In fact, the performance severely drops in other languages. Interestingly, Korean shows the lowest error rates which may be driven by the syllabic nature of the language, where the gap between Sylber 2.0 and text is narrower than others. Also the performance gap from Mimi is significant, which suggests our Sylber 2.0 embeddings are more easily mappable to text than previous SOTA speech tokens.
>
> To provide a more comprehensive view of information carried by Sylber 2.0, we evaluated the diverse speech tasks in the SUPERB benchmark.
>
> | Model       | PR PER ↓ | KS Acc ↑ | IC Acc ↑ | SID Acc ↑ | ER Acc ↑ | ASR WER ↓ | ASR (w/ LM) WER ↓ | QbE MTWV ↑ | SF F1 ↑ | SF CER ↓ | ASV EER ↓ | SD DER ↓ |
> |-------------|-----------|----------|----------|-----------|----------|-----------|---------------------|-------------|----------|-----------|------------|-----------|
> | Mimi        | 39.79     | 93.12    | 90.61    | 34.23     | 56.05    | 91.04     | 88.65               | 0.0108      | 6.72     | 98.86     | 12.17      | 13.37     |
> | Sylber      | 88.79     | **97.11**| **99.08**| 51.25     | **65.25**| **12.04** | **8.88**            | **0.0591**  | 85.66    | **29.49** | 8.75       | 15.55     |
> | Sylber 2.0  | **14.74** | 96.85    | 98.05    | **72.03** | 62.37    | 14.23     | 10.04               | 0.0471      | **85.94**| 30.09     | **5.69**   | **6.10**  |
>
> Sylber 2.0 shows on par or slightly worse performance than Sylber but shows significantly better performance in speaker-related tasks: SID, ASV, and SD. This is natural as we augmented Sylber with the acoustic encoder that can better inform acoustic characteristics of speakers. Being language universal is a potential source of performance degradation in other tasks as they are all in English, though the degree is marginal. In all metrics, Sylber 2.0 outperforms Mimi.

---

> ### Author Response · Authors · 2025-11-24
> **Response to Insufficient downstream task validation (2/2)**
>
> To sum, the new results on a broader range of downstream tasks suggest the effectiveness of Sylber 2.0 over the previous SOTA token, Mimi. Also, the original Sylber may be useful in the English only case while our new model can work language agnostically, which is well substantiated by the results in low-resource ASR.
>
> [1] Yang, S. W., Chi, P. H., Chuang, Y. S., Lai, C. I. J., Lakhotia, K., Lin, Y. Y., ... & Lee, H. Y. (2021). SUPERB: Speech Processing Universal PERformance Benchmark. Interspeech 2021.

---

> ### Author Response · Authors · 2025-11-24
> **Response to Overclaimed TTS efficiency benefits (1/2)**
>
> ### **2. Response to Overclaimed TTS efficiency benefits**
>
> We greatly appreciate the suggestions made by the reviewer. We have added an additional evaluation corpus by SeedTTS as suggested and added SparkTTS in the comparison. We agree that our initial claim on efficiency on training data was overclaimed to some extent. Moreover, we erroneously overestimated the sim-o score by using a different speaker encoder than the previous models. However, we still strongly believe Sylber 2.0 can enable parameter efficient generative speech models by being closer to syllables which are by definition the units of human speech production.
>
> To substantiate our claim, we conducted a controlled experiment using alternative target representation: Mel spectrogram or Mimi. We train the same model with the same resource settings as SylFlow to directly answer the reviewer’s question whether Sylber 2.0 is necessary over the Mel spectrogram. Instead of prompting and generating Sylber 2.0, the model takes in and generates Mel or Mimi, then the vocoder synthesizes the waveform. We use Mimi decoder, and HifiGAN from CosyVoice2 for Mel vocoder. (Please check Appendix A.2.2 in the revised manuscript for the implementation details.) The results are organized in the below table with suggested fixes. Moreover, we also train the model with an extended training data by adding 47k hours from Emilia.
>
> | Model                             | #Params | Training Data     | Libri WER ↓ | Libri SIM-o ↑ | Libri UTMOS ↑ | Seed WER ↓ | Seed SIM-o ↑ | Seed UTMOS ↑ |
> |-----------------------------------|---------|-------------------|-------------|----------------|----------------|-------------|----------------|----------------|
> | **Ground Truth**                  | –       | –                 | 2.47        | 0.69           | 4.09           | 2.14        | 0.73           | –              |
> | **CosyVoice**                     | 300M    | Multi-170k        | 3.59        | 0.66           | –              | 4.08        | 0.64           | –              |
> | **CosyVoice 2**                   | 500M    | Multi-170k        | 2.47        | 0.65           | **4.35**       | 2.57        | 0.65           | –              |
> | **FireRedTTS**                    | 580M    | Multi-248k        | 2.69        | 0.47           | –              | 3.82        | 0.46           | –              |
> | **MaskGCT**                       | 1048M   | Emilia-100k       | 2.72        | **0.69**       | 3.90           | 2.62        | 0.71           | –              |
> | **F5-TTS**                        | 300M    | Emilia-100k       | 2.42        | 0.66           | 3.88           | 1.83        | 0.65           | –              |
> | **DiTAR**                         | 600M    | Emilia-100k       | 2.39        | 0.67           | 4.22           | 1.69        | 0.74           | –              |
> | **SparkTTS**                      | 500M    | Multi-100k        | –           | –              | 4.35           | 1.98        | 0.58           | –              |
> | **CLEAR-Base**                    | 439M    | Libri-50k         | 2.21        | 0.59           | 4.22           | –           | –              | –              |
> | **CLEAR-Large**                   | 686M    | Libri-50k         | **1.88**    | 0.59           | 4.22           | –           | –              | –              |
> | **SylFlow (Ours)**                | 72M     | LibriTTS-0.6k     | 3.10        | 0.31           | 4.27           | 2.62        | 0.31           | 4.19           |
> |                                   |         | +Emilia-47k       | 2.35        | 0.36           | 4.33           | 1.92        | 0.35           | 4.31           |
> | *Ablation — Replacing Sylber 2.0* |         |                   |             |                |                |             |                |                |
> | ↳ *with Mel*                      | 109M    | LibriTTS-0.6k     | 5.73        | 0.18           | 3.28           | 4.48        | 0.18           | 3.38           |
> | ↳ *with Mimi*                     | 74M     | LibriTTS-0.6k     | 10.36       | 0.30           | 2.38           | 8.21        | 0.31           | 2.49           |
>
> First of all, using Mel and Mimi shows severe drops in performance in our controlled setting, where the drops are especially significant in WER and UTMOS. While the architecture may be suboptimal, those two baselines, Mel and Mimi, are proven to be effective in TTS, but only after using large models. At least, when constrained to our minimal setting, those baselines fail severely. This indicates that Sylber 2.0 has a better structure of information that can be more easily mappable from text, so that the small number of parameters is sufficient to generate accurate, high quality speech.

---

> ### Author Response · Authors · 2025-11-24
> **Response to Overclaimed TTS efficiency benefits (2/2)**
>
> We agree on the reviewer’s comment that a smaller dataset can fit in a smaller model size. Indeed, after fixing the speaker encoder, our small size model has a significant performance gap with the SOTA TTS models, indicating that our model is not well capturing/generating styles. However, our model still produces low WER and high UTMOS, and the gap from the SOTA model gets very narrower by scaling the training data but keeping the model size the same. While our small size of parameters may not be sufficient for cloning diverse speaking styles, this size is sufficient to generate highly accurate and high quality speech, with competitively low WER (rank 3rd in LibriSpeech-PC eval and rank 2nd in SeedTTS En) and high UTMOS. We believe this is still an impressive parameter efficiency which is never demonstrated in the field up to our knowledge. (We have modified our statement in the manuscript to be more accurate.)
>
> Nonetheless, the missing capacity of speaker cloning is crucial for zero-shot TTS in addition to intelligibility and quality, which may be resolved by scalining model parameters to a moderate level. Since our main focus is not at TTS, rather, in suggesting universal syllable embeddings, we leave this as a future direction as well as extending to different styles and languages. We believe the strong results in resynthesis in diverse styles and languages will support this idea.

---

> ### Author Response · Authors · 2025-11-24
> **Response to Unclear advantage over model-free representations**
>
> ### **3. Response to Unclear advantage over model-free representations**
>
> Indeed, any signals, even raw waveform, can be used for any speech model if a proper amount of training data and an appropriate architecture are supported. However, speech information has linguistic hierarchy beyond low-level acoustic information. Being closer to a higher level representation will allow a more efficient modeling of speech otherwise the downstream model should learn a mapping from low-level to the higher level or vice versa, which requires more parameters and training data. Sylber 2.0 provides a pretrained transformation to the higher-level speech information at syllable level. The high performance in resynthesis suggests that such transformation is generalizable to unseen contents, speakers, and styles. Our major contribution is that this is by far the highest level that is solely driven from speech without text. Moreover, we demonstrate that this can even be an achievable language agnostically. As a result, Sylber 2.0 embeddings can provide a short cut to textual representation. This is well supported by new experiments on TTS and ASR, and thus justifies the benefits over Mel spectrogram.

---

### Author Response · Authors · 2025-11-30
**General Response**

We sincerely appreciate the reviewers’ time and efforts for reviewing our papers. We have successfully addressed all of the points raised by reviewers in our responses. Since the discussions have been halted due to the incident and the ACs are newly assigned, we would like to summarize our contributions, initial reviews, and our responses.

In this work, we propose **the first universal syllabic speech coding framework** that can compress and abstract waveforms into syllabic embeddings with around **5Hz resolution, which is the lowest ever reported in the field**. Our framework, ***Sylber 2.0***, achieves this **without any supervision from text or human annotation** while enabling a near loss-less reconstruction back to the original waveform of **any languages and styles**. Our analyses suggest that these embeddings are well grounded in syllables, and by being grounded in the actual units (syllables) of human speech production, our model demonstrates **a stronger efficiency in downstream tasks** (TTS and ASR) than previous speech tokenization methods. Moreover, our work is highly conceptually novel as “universality” of syllabification has been controversial in linguistic literature for decades, where we can provide a full data-driven, emergent solution for this.

These contributions are well recognized by all of the reviewers. While we have covered a vast range of evaluation in speech reconstruction in the manuscript,  the reviewers commonly raised concerns on the shortage of downstream evaluation and ablation experiments. Therefore, during the rebuttal period, we have conducted an extensive range of downstream evaluation, ablation experiments, and additional analyses.

* New TTS experiments using baseline speech representations show severe drops in performance compared to using Sylber 2.0, confirming the superior efficiency of Sylber 2.0 in generative modeling (Section 6.1).
An additional TTS experiment with more data closes the gap with SOTA TTS models in terms of intelligibility and quality while maintaining 5-10 times smaller model size, which is consistent with an additional evaluation dataset (SeedTTS, suggested by reviewer QKpL) (Section 6.1).
* New downstream evaluation on multilingual low-resource ASR results demonstrate that Sylber 2.0 can be more beneficial in multilingual speech processing than previous speech tokens (Section 6.2).
* Additional downstream evaluation on SUPERB benchmark with 11 downstream tasks provide a wide range of potential downstream utilities of Sylber 2.0 (Appendix B.2).
* Ablation experiments show the necessity of each component of new acoustic encoder and position encoding proposed in Sylber 2.0 to ensure high-fidelity and generalizability (Appendix B.3).
* We provide a more detailed decomposition of the real-time factor (RTF) of Sylber 2.0 inference, compared to previous speech tokenization and encoding models (Appendix B.1), as reviewer F9nD requested.
* We visually inspected the embedding space of Sylber 2.0, which shows that the embeddings of Sylber 2.0 are phonetically organized while effectively disentangling linguistic contents from low-level acoustics (Appendix B.4).

Our new experiments can successfully address reviewers’ concerns on the limited coverage of the experiments, and more strongly substantiate our claims. Moreover, we have addressed the other individual concerns.

* The reviewer QKpL’s concern on unclear and overclaimed benefits over non-parametric acoustic features like Mel spectrogram. Our additional experiments in TTS and ASR show the clear advantages of Sylber 2.0 compared to Mel spectrogram.
* The reviewer F9nD and RWPH were worried about high WER (or low intelligibility) in TTS application, which may contradict with the fact * that Sylber 2.0 is syllabic. We can successfully resolve this issue by adding more training data but keeping the model size very small. As this additional data is comparable or less than the SOTA TTS models use, we can conclude the Sylber 2.0 can enable parameter-efficient generation of intelligible speech. This is also corroborated with a controlled comparison using baselines, Mel and Mimi, which show huge performance drops.
* As the reviewer F9nD requested, we provide a detailed analysis of the RTF which shows the fast inference time of Sylber 2.0 even with the additional architectures.
* The reviewer 3zrY’s question on whether Sylber 2.0 is linguistically meaningful can be answered by our qualitative analysis on the visualized embedding space, along with the effectiveness of Sylber 2.0 in the multilingual ASR tasks.
We have responded to the reviewer 3zrY’s concern on oversegmentation: it is induced by the system design, and a desirable property for universality across languages.

---

> ### Author Response · Authors · 2025-11-30
> **General Response (cont'd)**
>
> * The reviewer F9nD and RWPH were concerned with limited technical novelty. As stated above, the core novelty of our work is providing both conceptual and technical leap in understanding and modeling speech representation. Unlike dense frame tokenizations, we demonstrate that speech with any language or styles can be naturally structured in segments, providing interpretable handles of time-continuous acoustic signals and reducing token length at unprecedented levels. The previous works (Sylber and SyllableLM) have demonstrated this in English but provide significantly incomplete representations. And achieving language and style universality is non-trivial given the long-held controversy in linguistics regarding language-agnostic phonology. Moreover, the TTS using syllabic embedding has never been demonstrated, even though it is a natural task that mimics human speech production and syllables are by definition the units of speech production.
>
> We have posted more detailed responses to each individual reviewer with quantitative evidence from the new experiments. While not all reviewers have responded to our response, the reviewer RWPH has recognized that we have successfully addressed their concerns and raised their score (6→8) before the incident happened. We were expecting more raises from others with confidence since we have provided quantitative evidence with depth and breadth during the rebuttal period. We strongly believe that we have addressed all of the reviewer’s points, and we are happy to address any further questions if the discussion opens again.

---

### Meta-Review · Area_Chair_Hbvu · 2026-01-08

**Summary:**

There are several concerns among reviewers, including a lack of novelty (raised by reviewer F9nD and RWPH), unclear what the goal is (raised by reviewer QKpL), having too many confounding factors in the experiments (raised by RWPH), and finally whether the learned boundaries are linguistically meaningful (raised by reviewer 3zrY). I do agree with reviewer QKpL here that the authors are a little lost about what the actual goal is, and this can be evidenced by the TTS and ASR evaluation, until the reviewers asked for more downstream tasks. We can always evaluate on more benchmark, but this does not answer the fundamental question: what do we actually want from syllable embeddings. This is further evidenced in the rebuttal, where the emphasis is on achieving the lowest frame rate and being strong on both TTS and ASR, not on representing syllables.

**Reviewer Concerns:**

The rebuttal has successfully addressed the concerns on insufficient downstream tasks and missing ablations. However, the core concern remains unaddressed. Addressing this core concern is important because it's part of the reason why reviewer F9nD and RWPH think the novelty is lacking and why reviewer 3zrY is not convinced that the boundaries are linguistically meaningful.

**Reviewer Scores:**

Reviewer RWPH has explicitly agreed to raise the score, but reviewer RWPH's score is already leaning towards acceptance.

---

### Decision · Program_Chairs · 2026-01-26

Reject